# Smooth ECE: Principled Reliability Diagrams via Kernel Smoothing

**Jarosław Błasiok**
Columbia University

**Preetum Nakkiran**
Apple

## Abstract

Calibration measures and reliability diagrams are two fundamental tools for measuring and interpreting the calibration of probabilistic predictors. Calibration measures quantify the degree of miscalibration, and reliability diagrams visualize the structure of this miscalibration. However, the most common constructions of reliability diagrams and calibration measures — binning and ECE — both suffer from well-known flaws (e.g. discontinuity). We show that a simple modification fixes both constructions: first smooth the observations using an RBF kernel, then compute the Expected Calibration Error (ECE) of this smoothed function. We prove that with a careful choice of bandwidth, this method yields a calibration measure that is well-behaved in the sense of Błasiok, Gopalan, Hu, and Nakkiran (2023) — a *consistent calibration measure*. We call this measure the *SmoothECE*. Moreover, the reliability diagram obtained from this smoothed function visually encodes the SmoothECE, just as binned reliability diagrams encode the BinnedECE.

We also release a Python package with simple, hyperparameter-free methods for measuring and plotting calibration: `pip install relplot`. Code at: https://github.com/apple/ml-calibration.

## 1 Introduction

Calibration is a fundamental aspect of probabilistic predictors, capturing how well predicted probabilities of events match their true frequencies (Dawid, 1982). For example, a weather forecasting model is perfectly calibrated (also called "perfectly reliable") if among the days it predicts a 10% chance of rain, the observed frequency of rain is exactly 10%. There are two key questions in studying calibration: First, for a given predictive model, how do we measure its overall amount of miscalibration? This is useful for ranking different models by their reliability, and determining how much to trust a given model's predictions. Methods for quantifying miscalibration are known as *calibration measures*. Second, how do we convey *where* the miscalibration occurs? This is useful for better understanding an individual predictor's behavior (where it is likely to be over- vs. under-confident), as well as for re-calibration— modifying the predictor to make it better calibrated. The standard way to convey this information is known as a *reliability diagram*. Unfortunately, in machine learning, the most common methods of constructing both calibration measures and reliability diagrams suffer from well-known flaws, which we describe below.

The most common choice of calibration measure in machine learning is the Expected Calibration Error (ECE), more specifically its empirical variant the Binned ECE (Naeini et al., 2015). The ECE is known to be unsatisfactory for many reasons; for example, it is a discontinuous functional, so changing the predictor by an infinitesimally small amount may change its ECE drastically[1] (Kakade & Foster, 2008; Foster & Hart, 2018; Błasiok et al., 2023). Moreover, the ECE is impossible to estimate efficiently from samples (Lee et al., 2022; Arrieta-Ibarra et al., 2022), and its sample-efficient variant, the Binned ECE, is overly sensitive to choice of bin widths (Nixon et al., 2019; Kumar et al., 2019; Minderer et al., 2021). These shortcomings have been well-documented in the community, which motivated proposals of new, better-behaved calibration measures (e.g. Roelofs et al. (2022); Arrieta-Ibarra et al. (2022); Lee et al. (2022)).

---

[1] See Figure 3 and Appendix E for an illustration of this discontinuity.

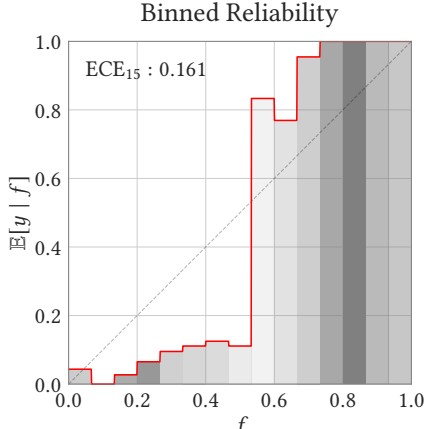 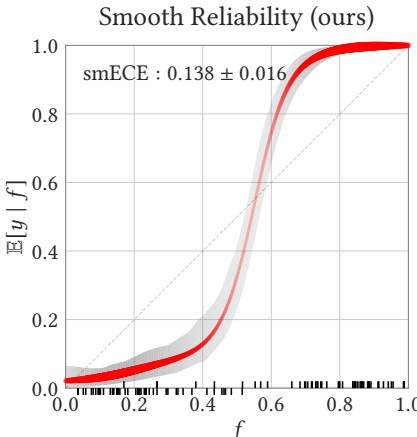

Figure 1: **Left:** Traditional reliability diagram based on binning, which is equivalent to histogram regression. **Right:** Proposed reliability diagram based on kernel regression, with our theoretically-justified choice of bandwidth. The width of the red line corresponds to the density of predictions, and the shaded region shows bootstrapped confidence intervals. Plot generated by our Python package.

Recently, Błasiok et al. (2023) proposed a theoretical definition of what constitutes a "good" calibration measure. The key principle is that good measures should provide upper and lower bounds on the calibration distance dCE, which is the Wasserstein distance between the joint distribution of prediction-outcome pairs, and the set of perfectly calibrated such distributions (formally defined in Definition 5 below). Calibration measures which satisfy this property are called *consistent calibration measures*. In light of this line of work, one may think that the question of which calibration measure to choose is largely resolved: simply pick a consistent calibration measure, such as Laplace Kernel Calibration Error / MMCE (Błasiok et al., 2023; Kumar et al., 2018), as suggested by Błasiok et al. (2023). However, this theoretical suggestion belies the practical reality: Binned ECE remains the most popular calibration measure used in practice, even in recent studies. We believe this is partly because Binned ECE enjoys an additional property: it can be visually represented by a specific kind of reliability diagram, namely the binned histogram. This raises the question of whether there are calibration measures which are *consistent* in the sense of Błasiok et al. (2023), and can also be represented by an appropriate reliability diagram. To be precise, we must discuss reliability diagrams more formally.

**Reliability Diagrams.** We consider measuring calibration in the setting of binary outcomes, for simplicity. Here, we have a joint distribution $(f, y) \sim \mathcal{D}$ over predictions $f \in [0, 1]$, and true outcomes $y \in \{0, 1\}$. We interpret $f$ as the predicted probability that $y = 1$. The "calibration function" $\mu : [0, 1] \to [0, 1]$ is defined as the conditional expectation:

$$\mu(f) := \mathbb{E}_{\mathcal{D}}[y \mid f].$$

A perfectly calibrated distribution, by definition, is one with a diagonal calibration function: $\mu(f) = f$. Reliability diagrams are traditionally thought of as estimates of the calibration function $\mu$ (Naeini et al., 2014; Bröcker, 2008). In other words, *reliability diagrams are one-dimensional regression methods*, since the goal of regressing $y$ on $f$ is exactly to estimate the regression function $\mathbb{E}[y \mid f]$. The practice of "binning" to construct reliability diagrams (as in Figure 1 left) can be equivalently thought of as using histogram regression to regress $y$ on $f$.

With this perspective on reliability diagrams, one may wonder why histogram regression is still the most popular method, when more sophisticated regressors are available. One potential answer is that users of reliability diagrams have an additional desiderata: it should be easy to visually read off a reasonable calibration measure from the reliability diagram. For example, it is easy to visually read off the Binned ECE from a binned reliability diagram, because it is simply the integrated absolute deviation from the diagonal:

$$\text{BinnedECE}_k = \int_0^1 \left| \hat{\mu}_k(f) - \overline{f}_k \right| dF$$

where $k$ is the number of bins, $\hat{\mu}_k$ is the histogram regression estimate of $y$ given $f$, and $\overline{f}_k$ is the "binned" version of $f$ — formally the histogram regression estimate of $f$ given $f$. This relationship is even more transparent for the full (non-binned) ECE, where we have

$$\text{ECE} = \int_0^1 |\mu(f) - f| \, dF = \mathbb{E}_f[|\mu(f) - f|]$$

where $\mu$ is the true regression function as above. However, more sophisticated regression methods do not neccesarily have such tight relationships to calibration measures. Thus we have a situation where better calibration measures exist, but they are not accompanied by reliability diagrams, and conversely better reliability diagrams exist (i.e. regression methods), but they are not associated with consistent calibration measures. We address this situation here: we present a new consistent calibration measure, *SmoothECE*, along with a regression method which naturally encodes this calibration measure. The SmoothECE is, per its name, equivalent to the ECE of a "smoothed" version of the original distribution, and the resulting reliability diagram can thus be interpreted as a smoothed estimate of the calibration function.

We emphasize that the idea of smoothing is not new — Gaussian kernel smoothing has been explicitly proposed as method for constructing reliability diagrams in the past (e.g. Bröcker (2008), as discussed in Arrieta-Ibarra et al. (2022)). Our contribution is two-fold: first, we give strong theoretical justification for kernel smoothing by proving it induces a consistent calibration measure. Second, and of more practical relevance, we show how to chose the kernel bandwidth in a principled way, which differs significantly from existing recommendations.

## 1.1 OVERVIEW OF METHOD

We start by describing the regression method, which defines our reliability diagram. We are given i.i.d. observations $\{(f_1, y_1), (f_2, y_2) \dots (f_k, y_k)\}$ where $f_i \in [0, 1]$ is the $i$-th prediction, and $y_i \in \{0, 1\}$ is the corresponding outcome. For example, if we are measuring calibration of an ML model on a dataset of validation samples, we will have $f_i = F(x_i)$ for model $F$ evaluated on sample $x_i$, with ground-truth label $y_i$. We would like to estimate the true calibration function $\mu(f) := \mathbb{E}[f \mid y]$. Our estimate $\hat{\mu}(f)$ is given by Nadaraya-Watson kernel regression (kernel smoothing) on this dataset (see Nadaraya (1964); Watson (1964); Simonoff (1996)):

$$\hat{\mu}(f) := \frac{\sum_i K_\sigma(f, f_i) y_i}{\sum_i K_\sigma(f, f_i)}. \tag{1}$$

That is, for a given $f \in [0, 1]$ our estimate of $y$ is the weighted average of all $y_i$, where weights are given by the kernel function $K_\sigma(f, f_i)$. The choice of kernel, and in particular the choice of bandwidth $\sigma$, is crucial for our method's theoretical guarantees. We use an essentially standard kernel (defined formally in Section 3): the Gaussian Kernel, reflected appropriately to handle boundary-effects of the interval $[0, 1]$. Our choice of bandwidth $\sigma$ is more subtle, but it is not a hyperparameter – we describe the explicit algorithm for choosing $\sigma$ in Section 3. It suffices to say for now that the amount of smoothing $\sigma$ will end up being proportional to the reported calibration error.

**Reliability Diagram**    We then construct a reliability diagram in the standard way, by displaying a plot of the estimated calibration function $\hat{\mu}$ along with a kernel density estimate of the predictions $f_i$ (see Figure 1). These two estimates, compactly presented on the same diagram, provide a tool to quickly understand and visually assess calibration properties of a given predictor. Moreover, they can be used to define a quantiative measure of overall degree of miscalibration, as we show below.

**SmoothECE**    A natural measure of calibration can be easily computed from data in the above reliability diagram. Specifically, let $\hat{\delta}(f)$ be the kernel density estimate of predictions: $\hat{\delta}(f) := \frac{1}{n} \sum_i K_\sigma(f, f_i)$. Then, similar to the definition of ECE, we can integrate the deviation of $\hat{\mu}$ from the diagonal to obtain:

$$\widetilde{\text{smECE}}_\sigma := \int |\hat{\mu}(t) - t| \hat{\delta}(t) dt.$$

The measure of calibration we actually propose in Section 3, $\mathsf{smECE}_\sigma$, is closely related but not identical to the above. Briefly, to define $\mathsf{smECE}_\sigma$ we consider the kernel smoothing of the difference between the outcome and the prediction $(y_i - f_i)$ instead of just smoothing the outcomes $y_i$. As it turns out, those choices lead to a calibration measure with better mathematical properties: $\mathsf{smECE}_\sigma$ is monotone decreasing as the kernel bandwidth $\sigma$ is increased, and $\mathsf{smECE}$, at our specific bandwidth choice and applied to the population distribution, is 0 for perfectly calibrated predictors.

We reiterate that the choice of the scale $\sigma$ is very important: too large or too small bandwidth will prevent the SmoothECE from being a consistent calibration measure. In Section 3, we will show how to algorithmically define the correct scale $\sigma^*$. For the reliability diagram, we suggest presenting the estimates $\hat{y}$ and $\hat{\delta}$ with the same scale $\sigma^*$, and for this scale we indeed have $\widetilde{\mathsf{smECE}}_{\sigma^*} \approx \mathsf{smECE}_{\sigma^*}$ (see Appendix A). Finally, note that we have been discussing finite-sample estimators of all quantities; the corresponding population quantities are defined analogously in Section 3.

### 1.2 SUMMARY OF OUR CONTRIBUTIONS

1. **SmoothECE.** We define a new hyperparmeter-free calibration measure, which we call the *SmoothECE* (abbreviated $\mathsf{smECE}$). We prove that the SmoothECE is a *consistent calibration measure*, in the sense of Błasiok et al. (2023). It also corresponds to a natural notion of distance: if SmoothECE is $\varepsilon$, then the function $f$ can be stochastically post-processed to make it perfectly calibrated, without perturbing $f$ by more than $\varepsilon$ in $L_1$.

2. **Smoothed Reliability Diagrams.** We show how to construct principled reliability diagrams which visually encode the SmoothECE. These diagrams can be thought of as "smoothed" versions of the usual binned reliability diagrams, where we perform Nadaraya-Watson kernel smoothing with the Gaussian kernel.

3. **Code.** We develop a Python package which computes the SmoothECE and plots the associated smooth reliability diagram (link omitted for anonymity). It is hyperparameter-free, efficient, and includes uncertainty quantification via bootstrapping. We include several experiments in Section 4, for demonstration purposes.

4. **Extensions to general metrics.** On the theoretical side, we investigate how far our construction of SmoothECE generalizes. We show that the notion of SmoothECE introduced in this paper can indeed be defined for a wider class of metrics on the space of predictions $[0, 1]$, and we prove the appropriate generalization of our main theorem: that the $\mathsf{smECE}$ for a given metric is a consistent calibration measure with respect to the same metric. Finally, perhaps surprisingly, we show that under specific conditions on the metric (which are satisfied, for instance, by the $d_{\text{logit}}$ metric), the associated $\mathsf{smECE}$ is in fact a consistent calibration measure *with respect to $\ell_1$ metric*.

**Organization.** We begin by discussing the closest related works (Section 2). In Section 3 we formally define the SmoothECE and prove its mathematical and computational properties. To aid intuition, we discuss the justification behind our various design choices in Appendix A. We then prove the SmoothECE can be estimated efficiently with respect to both sample-complexity and run-time (Section 3.4). Finally, we include experimental demonstrations of our method in Section 4, and conclude in Section 5. Extensions of our results to more general metrics are provided in Appendix B.

## 2 RELATED WORKS

**Reliability Diagrams and Binning.** Reliability diagrams, as far as we are aware, had their origins in the early reliability tables constructed by the meteorological community (Hallenbeck, 1920). These early accounts of calibration already applied the practice of binning— discretizing predictions into bins, and estimating frequencies conditional on each bin. Plots of these tables turned into binned reliability diagrams (Murphy & Winkler, 1977; DeGroot & Fienberg, 1983), which was recently popularized in the machine learning community by a series of works including Zadrozny & Elkan (2001); Niculescu-Mizil & Caruana (2005); Guo et al. (2017). Binned reliability diagrams continue to be used in studies of calibration in deep learning, including in the GPT-4 tech report (Guo et al., 2017; Nixon et al., 2019; Minderer et al., 2021; Desai & Durrett, 2020; OpenAI, 2023).

**Reliability Diagrams as Regression.** The connection between reliability diagrams and regression methods has been noted in the literature (e.g. Bröcker (2008); Copas (1983); Stephenson et al. (2008)). For example, Stephenson et al. (2008) observes "one can consider binning to be a crude form of non-parametric smoothing." However, we remind the reader of a subtlety: our objective in this work is *not* identical to the regression objective, since we want an estimator that is simultaneously a reasonable regression and a consistent calibration measure. Our choice of bandwidth must thus carefully balance the two; it cannot be simply be chosen to minimize the regression test loss.

**Alternate Constructions.** There have been various proposals to construct reliability diagrams which improve on binning; we mention several of them here. Many proposals can be seen as suggesting alternate regression techniques, to replace histogram regression. For example, some works suggest modifications to improve the binning method, such as adaptive bin widths or debiasing (Kumar et al., 2019; Nixon et al., 2019; Roelofs et al., 2022). These are closely related to data-dependent histogram estimators in the statistics literature (Nobel, 1996). Other works suggest using entirely different regression methods, including spline fitting (Gupta et al.), kernel smoothing (Bröcker, 2008; Popordanoska et al., 2022), and isotonic regression (Dimitriadis et al., 2021). The above methods for constructing regression-based reliability diagrams are closely related to methods for recalibration, since the ideal recalibration function is exactly the calibration function $\mu$. For example, isotonic regression (Barlow, 1972) has been used as both for recalibration (Zadrozny & Elkan, 2002; Naeini et al., 2015) and for reliability diagrams (Dimitriadis et al., 2021). Finally, Tygert (2020) and Arrieta-Ibarra et al. (2022) suggest visualizing reliability via cumulative-distribution plots, instead of estimating conditional expectations. While all the above proposals do improve upon binning in certain aspects, none of them ultimately induce consistent calibration measures in the sense of Błasiok et al. (2023). See Błasiok et al. (2023) for further discussion on the shortcomings of these measures.

**Multiclass Calibration.** We focus on binary calibration in this work. The multi-class setting introduces several new complications— foremost, there is no consensus on how best to define calibration measures in the multi-class setting; this is an active area of research (e.g. Vaicenavicius et al. (2019); Kull et al. (2019); Widmann et al. (2020)). However, our methods apply directly to any of the metrics induced by the "multiclass-to-binary" reductions of Gupta & Ramdas (2021), because these are determined by the standard calibration of a related binary problem. This includes, for example, the multi-class confidence calibration.

**Consistent Calibration Measures.** We warn the reader that the terminology of *consistent calibration measure* does not refer to the concept of statistical consistency. Rather, it refers to the definition introduced in Błasiok et al. (2023), to capture calibration measures which polynomially approximate the true (Wasserstein) distance to perfect calibration.

## 3 SMOOTH ECE

In this section we will define the calibration measure smECE. As it turns out, it has slightly better mathematical properties than $\widetilde{\mathsf{smECE}}$ defined in Section 1.1, and those properties will allow us to chose the proper scale $\sigma$ in a more principled way — moreover, we will be able to relate smECE with $\widetilde{\mathsf{smECE}}$.

Specifically, the measure $\mathsf{smECE}_\sigma$ enjoys the following convenient mathematical properties, which we will prove in this section.

- The $\mathsf{smECE}_\sigma(\mathcal{D})$ is monotone decreasing as we increase the smoothing parameter $\sigma$.

- If $\mathcal{D}$ is perfectly calibrated distribution, then for any $\sigma$ we have $\mathsf{smECE}_\sigma(\mathcal{D}) = 0$. Indeed, for any $\sigma$ we have $\mathsf{smECE}_\sigma(\mathcal{D}) \leq \mathsf{ECE}(\mathcal{D})$.

- The $\mathsf{smECE}_\sigma$ is Lipschitz with respect to the Wasserstein distance on the space of distributions over $[0,1] \times \{0,1\}$: for any $\mathcal{D}_1, \mathcal{D}_2$ we have $|\mathsf{smECE}_\sigma(\mathcal{D}_1) - \mathsf{smECE}_\sigma(\mathcal{D}_2)| \leq (1 + \sigma^{-1})W_1(\mathcal{D}_1, \mathcal{D}_2)$. This implies $\mathsf{smECE}_\sigma(\mathcal{D}) \leq (1 + \sigma^{-1})\underline{\mathrm{dCE}}(\mathcal{D})$.

- For any distribution $\mathcal{D}$ and any $\sigma$, there is a (probabilistic) post-processing $\kappa$, such that if $(f, y) \sim \mathcal{D}$, then the distribution $\mathcal{D}'$ of $(\kappa(f), y)$ is perfectly calibrated, and moreover $\mathbb{E}|f - \kappa(f)| \leq \mathsf{smECE}_\sigma(\mathcal{D}) + \sigma$. In particular $\underline{\mathrm{dCE}}(\mathcal{D}) \leq \mathsf{smECE}_\sigma + \sigma$.

**Reflected Gaussian Kernel.** In all of our kernel applications, we use a "reflected" version of the Gaussian kernel defined as follows. Let $\pi_R : \mathbb{R} \to [0, 1]$ be the projection function which is identity on $[0, 1]$, and collapses two points iff they differ by a composition of reflections around integers. That is $\pi_R(x) := (x \mod 2)$ if $(x \mod 2) \leq 1$, and $(2 - (x \mod 2))$ otherwise. The Reflected Gaussian kernel on $[0, 1]$ with scale $\sigma$, is then given by

$$\tilde{K}_\sigma(x, y) := \sum_{\tilde{x} \in \pi_R^{-1}(x)} \phi_\sigma(\tilde{x} - y) = \sum_{\tilde{y} \in \pi_R^{-1}(y)} \phi_\sigma(x - \tilde{y}), \tag{2}$$

where $\phi$ is the probability density function of $\mathcal{N}(0, 1)$, that is $\phi_\sigma(t) = \exp(-t^2/2\sigma^2)/\sqrt{2\pi\sigma^2}$. We use the Reflected Gaussian kernel in order to alleviate the bias introduced by standard Gaussian kernel near the boundaries of the interval $[0, 1]$.

### 3.1 DEFINING SMECE$_\sigma$ AT SCALE $\sigma$

We now present the construction of $\mathsf{smECE}_\sigma$, at a *given scale* $\sigma > 0$. We will show how to pick this $\sigma$ in the subsequent section. Let $\mathcal{D}$ be a distribution over $[0, 1] \times \{0, 1\}$ of the pair of prediction $f \in [0, 1]$ and outcome $y \in \{0, 1\}$. For a given kernel $K : \mathbb{R} \to \mathbb{R}$ we define the kernel smoothing of the residual $r := y - f$ as

$$\hat{r}_{\mathcal{D},K}(t) := \frac{\mathbb{E}_{(f,y)\sim\mathcal{D}} K(t, f)(y - f)}{\mathbb{E}_{(f,y)\sim\mathcal{D}} K(t, f)}. \tag{3}$$

This differs from the definition in Section 1.1, where we applied the kernel smoothing to the outcomes $y$ instead. In many cases of interest, the probability distribution $\mathcal{D}$ is going to be an empirical probability distribution over finite set of pairs $\{(f_i, y_i)\}$ of observed predictions $f_i$ and associated observed outcomes $y_i$. In this case, the $\hat{r}_{\mathcal{D}}(t)$ is just a weighted average of residuals $(f_i - y_i)$ where the weight of a given sample is determined by the kernel $K(f_i, t)$. This is equivalent to the Nadaraya-Watson kernel regression (a.k.a. kernel smoothing, see Nadaraya (1964); Watson (1964); Simonoff (1996)), estimating $(y - f)$ with respect to the independent variable $f$.

We consider also the kernel density estimation $\hat{\delta}_{\mathcal{D},K}(t) := \mathbb{E}_{f,y\sim\mathcal{D}} K(t, f)$. The $\mathsf{smECE}_K(\mathcal{D})$ is now defined as

$$\mathsf{smECE}_K(\mathcal{D}) := \int |\hat{r}_{\mathcal{D},K}(t)|\hat{\delta}_{\mathcal{D},K}(t) \, dt. \tag{4}$$

This notion is close to $\mathsf{ECE}$ of a smoothed distribution of $(f, y)$, in a sense which can be formalized (see Appendix A). For now, let us discuss the intuitive connection. For any distribution of prediction, and outcome $(f, y)$, we can consider an expected residual $r(t) := \mathbb{E}[f - y|f = t]$, then

$$\mathsf{ECE}(f, y) := \int |r(t)| \, d\mu_f(t),$$

where $\mu_f$ is a measure of $f$. We can compare this with (4), where the conditional residual $r$ has been replaced by its smoothed version $\hat{r}$, and the measure $\mu_f$ has been replaced by $\hat{\delta} \, dt$ – the measure of $f + \eta$ for some noise $\eta$. Equation (4) can be simplified by using the definitions of $\hat{\delta}_{\mathcal{D},K}$ and $\hat{r}_{\mathcal{D},K}$,

$$\mathsf{smECE}_K(\mathcal{D}) = \int \left| \mathbb{E}_{f,y} K(t, f)(y - f) \right| \, dt. \tag{5}$$

In what follows, we will be focusing on the reflected Gaussian kernel with scale $\sigma$, $\tilde{K}_{N,\sigma}$, and we shall use shorthand $\mathsf{smECE}_\sigma$ to denote $\mathsf{smECE}_{\tilde{K}_{N,\sigma}}$. We will now show how the scale $\sigma$ is chosen.

### 3.2 DEFINING SMECE: PROPER CHOICE OF SCALE

First, we observe that $\mathsf{smECE}_\sigma$ satisfies a natural monotonicity property: increasing the smoothing scale $\sigma$ decreases the $\mathsf{smECE}_\sigma$. (Proof of this and subsequent lemmas can be found in Appendix D.)

**Lemma 1.** *For a distribution $\mathcal{D}$ over $[0, 1] \times \{0, 1\}$ and $\sigma_1 \leq \sigma_2$, we have*

$$smECE_{\sigma_1}(\mathcal{D}) \geq smECE_{\sigma_2}(\mathcal{D}).$$

Several of our design choices were crucial to ensure this property: the choice of the reflected Gaussian kernel, and the choice to smooth the residual $(y - f)$ as opposed to the outcome $y$. Note that since $\mathsf{smECE}_\sigma(\mathcal{D}) \in [0, 1]$, and for a given predictor $\mathcal{D}$, the function $\sigma \mapsto \mathsf{smECE}_\sigma(\mathcal{D})$ is a non-increasing function of $\sigma$, there is a unique $\sigma^*$ s.t. $\mathsf{smECE}_{\sigma^*}(\mathcal{D}) = \sigma^*$ (and we can find it efficiently using binary search). Thus we can define:

**Definition 2** (SmoothECE). *We define $\mathsf{smECE}(\mathcal{D})$ to be the unique $\sigma^*$ satisfying $\mathsf{smECE}_{\sigma^*}(\mathcal{D}) = \sigma^*$. We also write this quantity as $\mathsf{smECE}_*(\mathcal{D})$ for clarity.*

### 3.3 smECE IS A CONSISTENT CALIBRATION MEASURE

We will show that $\sigma_*$ defined in the previous subsection is a convenient scale on which the $\mathsf{smECE}$ of $\mathcal{D}$ should be evaluated. The formal requirement that $\mathsf{smECE}_{\sigma^*}$ meets is captured by the notion of *consistent calibration measure*, introduced in Błasiok et al. (2023). We provide the definition below, but before we do, let us recall the definition of the *Wasserstein metric*.

For a metric space $(\mathcal{X}, d)$, let us consider $\Delta(\mathcal{X})$ to be the space of all probability distributions over $\mathcal{X}$. We define the *Wasserstein* metric on the space $\Delta(X)$ (sometimes called *earth-movers distance*) Peyré et al. (2019).

**Definition 3** (Wasserstein distance). *For two distributions $\mu, \nu \in \Delta(\mathcal{X})$ we define the Wasserstein distance*

$$W_1(\mu, \nu) := \inf_{\gamma \in \Gamma(\mu, \nu)} \mathbb{E}_{(x,y) \sim \gamma} d(x, y),$$

*where $\Gamma(\mu, \nu)$ is the family of all couplings of distributions $\mu$ and $\nu$.*

**Definition 4.** *A probability distribution $\mathcal{D}$ over $[0, 1] \times \{0, 1\}$ of prediction $f$ and outcome $y$ is perfectly calibrated if $\mathbb{E}_\mathcal{D}[y|f] = f$. We denote the family of all perfectly calibrated distributions by $\mathcal{P} \subset \Delta([0, 1] \times \{0, 1\})$.*

**Definition 5** (Consistent calibration measure (Błasiok et al., 2023)). *For a probability distribution $\mathcal{D}$ over $[0, 1] \times \{0, 1\}$ we define the distance to calibration $\underline{\mathrm{dCE}}(\mathcal{D})$ to be the Wasserstein distance to the nearest perfectly calibrated distribution, associated with metric $d$ on $[0, 1] \times \{0, 1\}$ which puts two disjoint intervals infinitely far from each other.*

*Concretely*

$$d((f_1, y_1), (f_2, y_2)) := \begin{cases} |f_1 - f_2| & \text{if } y_1 = y_2 \\ \infty & \text{otherwise} \end{cases}.$$

*and*

$$\underline{\mathrm{dCE}}(\mathcal{D}) = \inf_{\mathcal{D} \in \mathcal{P}} W_1(\mathcal{D}, \mathcal{D}').$$

*Finally, any function $\mu$ assigning to distributions over $[0, 1] \times \{0, 1\}$ a non-negative real calibration score, is called* consistent calibration measure *if it is polynomially upper and lower bounded by $\underline{\mathrm{dCE}}$, i.e. there are constants $c_1, c_2, \alpha_1, \alpha_2$, s.t.*

$$c_1 \underline{\mathrm{dCE}}(\mathcal{D})^{\alpha_1} \leq \mu(\mathcal{D}) \leq c_2 \underline{\mathrm{dCE}}(\mathcal{D})^{\alpha_2}.$$

With this definition in hand, we prove the following.

**Theorem 6.** *The measure $\mathsf{smECE}(\mathcal{D})$ is a consistent calibration measure.*

This theorem is a consequence of the following two inequalities. First of all, if we add the penalty proportional to the scale of noise $\sigma$, then $\mathsf{smECE}_\sigma$ upper bounds the distance to calibration.

**Lemma 7.** *For any $\sigma$, we have*

$$\underline{\mathrm{dCE}}(\mathcal{D}) \lesssim \mathsf{smECE}_\sigma(\mathcal{D}) + \sigma.$$

On the other hand, as soon as the scale of the noise is sufficiently large compared to the distance to calibration, the $\mathsf{smECE}$ of a predictor is itself upper bounded as follows.

**Lemma 8.** *Let $(f, y)$ be any predictor. Then for any $\sigma$ we have*

$$\mathsf{smECE}_\sigma(\mathcal{D}) \leq \left(1 + \frac{1}{\sigma}\right) \underline{\mathrm{dCE}}(\mathcal{D}).$$

*In particular if $\sigma > 2\sqrt{\underline{\mathrm{dCE}}(\mathcal{D})}$, then*

$$\mathsf{smECE}_\sigma(\mathcal{D}) \leq 2\sqrt{\underline{\mathrm{dCE}}(\mathcal{D})}.$$

This lemma, together with the fact that $\sigma \mapsto \mathsf{smECE}_\sigma$ is non-increasing, directly implies that the fixpoint satisfies $\sigma^* \leq 2\sqrt{\underline{\mathrm{dCE}}(\mathcal{D})}$. On the other hand, using Lemma 7, at this fixpoint we have $\underline{\mathrm{dCE}}(\mathcal{D}) \leq \sigma^* + \mathsf{smECE}_{\sigma^*}(\mathcal{D}) = 2\sigma^*$. That is

$$\frac{1}{2}\underline{\mathrm{dCE}}(\mathcal{D}) \leq \mathsf{smECE}_*(\mathcal{D}) \leq 2\sqrt{\underline{\mathrm{dCE}}(\mathcal{D})},$$

proving the Theorem 6.

## 3.4 SAMPLE AND RUNTIME EFFICIENCY

Here we prove that our method is efficient with respect to both sample complexity and runtime. We want to estimate $\mathsf{smECE}$ of the underlying distribution $\mathcal{D}$ over $[0,1] \times \{0,1\}$, using samples from this distribution. Specifically, let us sample independently at random $m$ pairs $(f_i, y_i) \sim \mathcal{D}$, and let us define $\hat{\mathcal{D}}$ to be the empirical distribution over the multiset $\{(f_i, y_i)\}$; that is, to sample from $\hat{\mathcal{D}}$, we pick a uniformly random $i \in [m]$ and output $(f_i, y_i)$. Then, the plug-in estimator satisfies the following generalization bound.

**Theorem 9.** *For a given $\sigma_0 > 0$ if $m \gtrsim \sigma_0^{-1}\varepsilon^{-2}$, then with probability at least $2/3$ over the choice of independent random sample $(f_i, y_i)_{i=1}^m$ (with $(f_i, y_i) \sim \mathcal{D}$), we have simultaneously for all $\sigma \geq \sigma_0$,*

$$|\mathsf{smECE}_\sigma(\mathcal{D}) - \mathsf{smECE}_\sigma(\hat{\mathcal{D}})| \leq \varepsilon.$$

*In particular if $\mathsf{smECE}_*(\mathcal{D}) > \sigma_0$, then (with probability at least $2/3$) we have $|\mathsf{smECE}_*(\mathcal{D}) - \mathsf{smECE}_*(\hat{\mathcal{D}})| \leq \varepsilon$.*

The proof is in Appendix D.6. The success probability $2/3$ can be amplified arbitrarily in the standard way, by taking the median of independent trials. Finally, it is clear that $\mathsf{smECE}$ can be computed efficiently, by using the Fast Fourier Transform for convolutions. For completeness, in Appendix C we show that $\mathsf{smECE}$ can be approximated up to error $\varepsilon$ in time $\mathcal{O}(n\log\varepsilon^{-1} + M\log\varepsilon^{-1}\log^{3/2}M)$ in the RAM model, where $M = \lceil\varepsilon^{-1}\sigma^{-1}\rceil$.

## 4 EXPERIMENTS

We include several experiments demonstrating our method on public datasets in various domains, from deep learning to meteorology. The sample sizes vary between several hundred to 50K, to show how our method behaves for different data sizes. In each setting, we compare the classical binned reliability diagram to the smooth diagram generated by our Python package. Our diagrams include kernel density estimates of the predictions (at the same kernel bandwidth $\sigma^*$ used to compute the SmoothECE). For binned diagrams, the number of bins is chosen to be optimal for the regression test MSE loss, optimized via cross-validation.

**Deep Networks.** Figure 2a shows the confidence calibration of ResNet32 (He et al., 2016) on the ImageNet validation set (Deng et al., 2009). ImageNet is an image classification task with 1000 classes, and has a validation set of 50,000 samples. In this multi-class setting, the model $f$ outputs a distribution over $k = 1000$ classes, $f : \mathcal{X} \to \Delta_k$. Confidence calibration is defined as calibration of the pairs $(\mathrm{argmax}_{c \in [k]} f_c(x)\ ,\ \mathbb{1}\{f(x) = y\})$, which is a distribution over $[0,1] \times \{0,1\}$. That is, confidence calibration measures the agreement between confidence and correctness of the predictions. We use the publicly available data from Hollemans (2020), evaluating the models trained by Wightman (2019).

**Solar Flares.** Figure 2b shows the calibration of a method for forecasting solar flares, over a period of 731 days. We use the data from Leka et al. (2019), which was used to compare reliability diagrams in Dimitriadis et al. (2021). Specifically, we consider forecasts of the event that a class

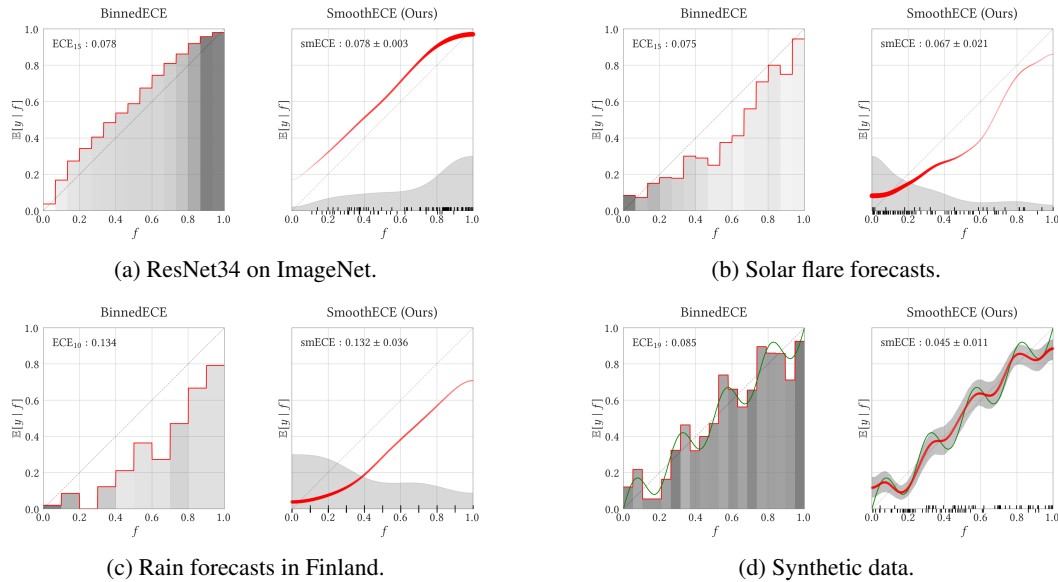

Figure 2: Comparison of binned reliability diagrams and our proposed smooth reliability diagrams, on several demonstration datasets.

C1.0+ solar flare occurs on a given day, made by the DAFFS forecasting model developed by North-West Research Associates. Overall, such solar flares occur on 25.7% of the 731 recorded days. We use the preprocesssed data from the replication code at: `https://github.com/TimoDimi/replication_DGJ20`. For further details of this dataset, we refer the reader to Dimitriadis et al. (2023, Section 6.1) and Leka et al. (2019).

**Precipitation in Finland.** Figure 2c shows the calibration of daily rain forecasts made by the Finnish Meteorological Institute (FMI) in 2003, for the city of Tampere in Finland. Forecasts are made for the probability that precipitation exceeds $0.2$mm over a 24 hour period; the dataset includes records for 346 days (Nurmi, 2003).

**Synthetic Data.** For demonstration purposes, we apply our method to a simple synthetic dataset in Figure 2d. One thousand samples $f_i \in [0,1]$ are drawn uniformly in the interval $[0,1]$, and the conditional distribution of labels $\mathbb{E}[y_i \mid f_i]$ is given by the green line in Figure 2d. Here, instead of kernel density estimates, we show bootstrapped confidence bands around our estimated regressor. Note that the true conditional distribution is non-monotonic in this example.

**Limitations.** One limitation of our method is that since it is generic, there may be better tools to use in special cases, when we can assume more structure in the prediction distributions. For example, if the predictor is known to only output a small finite set of possible probabilities, then it is reasonable to simple estimate conditional probabilities by using these points as individual bins. The rain forecasts in Figure 2c have this structure, since the forecasters only predict probabilities in multiples of 10% – in such cases, using bins which are correctly aligned is a very reasonable option.

## 5 CONCLUSION

We have presented a method of computing calibration error which is both mathematically well-behaved (i.e. *consistent* in the sense of Błasiok et al. (2023)), and can be visually represented in a reliability diagram. We also released a python package which efficiently implements our suggested method. We hope this work aids practitioners in computing, analyzing, and visualizing the reliability of probabilistic predictors.

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

## A    DISCUSSION: DESIGN CHOICES

Here we discuss the motivation behind several design choices that may a-priori seem ad-hoc. In particular, the choice to smooth the *residuals* $(y_i - f_i)$ when computing the smECE, but to smooth the outcomes $y_i$ directly when plotting the reliability diagram.

For the purpose of constructing the reliability diagram, it might be tempting to plot a function $y'(f) := \hat{r}(f) + f$ (of smoothed residual as defined in (3), shifted back by the prediction $f$), as well as the smoothed density $\hat{\delta}(t)$, as in the definition of smECE. This is a fairly reasonable approach, unfortunately it has a particularly undesirable feature — there is no reason for $y'(t) := \hat{r}(t) + t$ to be bounded in the interval $[0, 1]$. It is therefore visually quite counter-intuitive, as the plot of $y(t)$ is supposed to be related with our guess on the average outcome $y$ given (slightly noisy version of) the prediction $t$.

As discussed in Section 1.1, we instead consider the kernel regression on $y$, as opposed to the kernel regression on the residual $y - f$, and plot exactly this, together with the density $\hat{\delta}$. Specifically, let us define

$$\hat{y}_{\mathcal{D},K}(t) := \frac{\mathbb{E}_{f,y\sim\mathcal{D}} K(t,f)y}{\mathbb{E}_{f,y\sim\mathcal{D}} K(t,f)}. \tag{6}$$

and chose as the reliability diagram a plot of a pair of functions $t \mapsto \hat{y}_{\mathcal{D},K}(t)$ and $t \mapsto \hat{\delta}_{\mathcal{D},K}(t)$ — the first plot is our estimation (based on the kernel regression) of the outcome $y$ for a given prediction $t$, the other is the estimation of the density of prediction $t$. As discussed in the Section 1.1, we will focus specifically on the kernel $K$ being the reflected Gaussian kernel, defined by (2).

It is now tempting to define the calibration error related with this diagram, as an ECE of this new random variable over $[0,1] \times \{0,1\}$, analogously to the definition of smECE, by considering

$$\widetilde{\mathsf{smECE}}_\sigma(\mathcal{D}) := \int |\hat{y}_{\mathcal{D},K}(t) - t|\hat{\delta}_{\mathcal{D},K}(t)\,\mathrm{d}t. \tag{7}$$

This definition can be readily interpreted: for a random pair $(f,y)$ and an $\eta \sim \mathcal{N}(0,\sigma)$ independent, we can consider a pair $(f+\eta, y)$. It turns out that

$$\widetilde{\mathsf{smECE}}_\sigma(\mathcal{D}) = \mathsf{ECE}(\pi_R(f+\eta),y),$$

where $\pi_R : \mathbb{R} \to [0,1]$ collapses points that differ by reflection around integers (see Section 1.1).

Unfortunately, despite being directly connected with more desirable reliability diagrams, and having more immediate interpretation as a ECE of a noisy prediction, this newly introduced measure $\widetilde{\mathsf{smECE}}$ has its own problems, and is generally mathematically much poorer-behaved than smECE. In particular it is no longer the case that if we start with the perfectly calibrated distribution, and apply some smoothing with relatively large bandwidth $\sigma$, the value of the integral (7) stays small. In fact it might be growing as we add more smoothing[2].

Nevertheless, if we chose the correct bandwidth $\sigma^*$, as guided by the smECE consideration, the integral (7), which is visually encoded by the reliability diagram we propose, should still be within constant factor from the actual $\mathsf{smECE}_\sigma^*(\mathcal{D})$, and hence provides a consistent calibration measure

**Lemma 10.** *For any $\sigma$ we have*

$$\widetilde{\mathsf{smECE}}_\sigma(\mathcal{D}) = \mathsf{smECE}_\sigma(\mathcal{D}) \pm c\sigma,$$

*where $c = \sqrt{2/\pi} \le 0.8$.*

*In particular, for $\sigma^*$ s.t. $\mathsf{smECE}_{\sigma^*}(\mathcal{D}) = \sigma^*$, we have*

$$\widetilde{\mathsf{smECE}}_{\sigma^*}(\mathcal{D}) \approx \mathsf{smECE}_{\sigma^*}(\mathcal{D}).$$

(The proof can be found in Appendix D.3).

## B    GENERAL METRICS

Our previous discussion implicitly assumed the the trivial metric on the interval $d(u,v) = |u-v|$. We will now explore which aspects of our results extend to more general metrics over the interval $[0,1]$. This is relevant if, for example, our application downstream of the predictor is more sensitive to miscalibration near the boundaries.

The study of consistency measures with respect to general metrics is also motivated by the results of Błasiok et al. (2023). There it was shown that for any proper loss function $l$, there was an associated metric $d_l$ on $[0,1]$ such that the predictor has small weak calibration error with respect to $d_l$ if and only if the loss $l$ cannot be significantly improved by post-composition with a Lipschitz function with respect to $d_l$. Specifically, they proved

$$\mathrm{wCE}_{d_l}(\mathcal{D}) \lesssim \mathbb{E}_{(f,y)\sim\mathcal{D}}[l(f,y)] - \inf_\kappa \mathbb{E}_{(f,y)\sim\mathcal{D}}[l(\kappa(f),y)] \lesssim \sqrt{\mathrm{wCE}_{d_l}(\mathcal{D})},$$

---

[2]This can be easily seen, if we consider the trivial perfectly calibrated distribution, where outcome $y \sim$ Bernoulli$(1/2)$ and prediction $f$ is deterministic $1/2$. Then $\widetilde{\mathsf{smECE}}_\sigma(\mathcal{D}) = C\sigma$ for some constant $C \approx 0.79$.

where $\kappa : [0,1] \to [0,1]$ ranges over all functions Lipschitz with respect to the $d_l$ metric, and $\mathrm{wCE}_{d_l}(\mathcal{D})$ is the weak calibration error (as introduced by Kakade & Foster (2008), and extended to general metrics in Błasiok et al. (2023), see Definition 11).

The most intuitive special case of the above result is the square loss function, which corresponds to a trivial metric on the interval $d(u,v) = |u - v|$. In practice, different proper loss functions are also extensively used — the prime example being the *cross entropy loss* $l(f,y) := -(y \ln p + (1 - y) \ln(1-p))$, which is connected with the metric $d_{logit}(u,v) := |\log(u/(1-u)) - \log(v/(1-v))|$ on $[0,1]$. Thus, we may want to generalize our results to also apply to non-trivial metrics.

## B.1   GENERAL DUALITY

We will prove a more general statement of the duality theorem in Błasiok et al. (2023). Specifically, they showed that the minimization problem in the definition of the dCE, can be dualy expressed as a maximal correlation between residual $r := y - f$ and a bounded Lipschitz function of the prediction $f$. This notion, which we will refer to as *weak calibration error* first appeared in Kakade & Foster (2008), and was further explored in Gopalan et al. (2022); Błasiok et al. (2023); Błasiok et al. (2023)[3].

**Definition 11** (Weak calibration error). *For a distribution $\mathcal{D}$ over $[0,1] \times \{0,1\}$ of pairs of prediction and outcome, and a metric $d$ on the space $[0,1]$ of all possible predictions, we define*

$$\mathrm{wCE}_d(\mathcal{D}) := \sup_{w \in \mathcal{L}_d} \mathbb{E}_{f,y\sim\mathcal{D}} (f - y)w(f), \tag{8}$$

*where the supremum is taken over all functions $w : [0,1] \to [-1,1]$ which are 1-Lipschitz with respect to the metric $d$.*[4]

For the trivial metric on the interval $d(u,v) = |u-v|$, wCE was known to be linearly related with dCE by Błasiok et al. (2023). We show in this paper that the duality theorem connecting wCE and dCE holds much more generally, for a broad family of metrics.

**Theorem 12** (Błasiok et al. (2023)). *If a metric $d$ on the interval satisfies $d(u,v) \gtrsim |v - u|$ then $\mathrm{wCE}_d \approx \underline{\mathrm{dCE}}_d$.*

The more general formulation provided by Theorem 12 can be shown by following closely the original proof step by step. We provide an alternate proof (simplified and streamlined) in Appendix D.8.

## B.2   THE $\underline{\mathrm{dCE}}_{d_{logit}}$ IS A CONSISTENT CALIBRATION MEASURE WITH RESPECT TO $\ell_1$

As in turns out, for a relatively wide family of metrics on the space of predictions (including the $d_{logit}$ metric), the associated calibration measures are consistent calibration measures *with respect to the $\ell_1$ metric*. The main theorem we prove in this section is the following.

**Theorem 13.** *If a metric $d : [0,1]^2 \to \mathbb{R} \cup \{\pm\infty\}$ satisfies $d(u,v) \gtrsim |u-v|$ and moreover for some $c > 0$,*

$$\forall \varepsilon, \ \forall u,v \in [\varepsilon, 1-\varepsilon], \quad d(u,v) \le |u-v|\varepsilon^{-c},$$

*then $\underline{\mathrm{dCE}}_d$ is a consistent calibration measure.*

The proof of this theorem (as is the case for many proofs of consistency for calibration measures) heavily uses the duality Theorem 12 — since proving that a function is a consistent calibration measure amounts to providing a lower and upper bound, it is often convenient to use the dCE formulation for one bound and wCE for the other.

The lower bound in Theorem 13 is immediate — since $d(u,v) \ge \ell_1(u,v)$, the induced Wasserstein distances on the space $[0,1] \times \{0,1\}$ satisfy the same inequality, hence $\underline{\mathrm{dCE}}_d \ge \underline{\mathrm{dCE}}_{\ell_1}$, and $\underline{\mathrm{dCE}}_{\ell_1} \ge \underline{\mathrm{dCE}}/2$ by Claim 31.

As it turns out, if the metric of interest is well-behaved except near the endpoints of the unit interval, we can also prove the converse inequality, and lower bound $\mathrm{wCE}(\mathcal{D})$ by polynomial of $\mathrm{wCE}_d(\mathcal{D})$.

---

[3]Weak calibration was called *smooth calibration error* in Gopalan et al. (2022); Błasiok et al. (2023). We revert back to the original terminology *weak calibration error* to avoid confusion with the notion of smECE developed in this paper.

[4]In Błasiok et al. (2023)

**Lemma 14.** *Let $d : [0,1]^2 \to \mathbb{R}_+ \cup \{\infty\}$ be any metric satisfying for some $c > 0$,*

$$\forall \varepsilon, \ \forall u, v \in [\varepsilon, 1-\varepsilon], \quad d(u,v) \le |u-v|\varepsilon^{-c},$$

*then $\mathrm{wCE}_d(\mathcal{D})^q \lesssim \mathrm{wCE}(\mathcal{D})$, where $q := \max(c+1, 2)$.*

(Proof in Appendix D.5.)

We are ready now to prove the main theorem here

*Proof of Theorem 13.* We have $\underline{\mathrm{dCE}}_d(\mathcal{D}) \ge \underline{\mathrm{dCE}}(\mathcal{D})/2$ by our previous discussion, on the other hand Theorem 12 and Lemma 14 imply the converse inequality:

$$\underline{\mathrm{dCE}}_d(\mathcal{D}) \approx \mathrm{wCE}_d(\mathcal{D}) \le \mathrm{wCE}(\mathcal{D})^{1/q} \approx \underline{\mathrm{dCE}}(\mathcal{D})^{1/q}.$$

$\square$

**Corollary 15.** *For a metric induced by cross-entropy loss function $d_{logit}(u,v) := |\ln(u/(1-v)) - \ln(v/(1-v))|$, the $\mathrm{wCE}_{d_{logit}}$ is a consistent calibration measure.*

*Proof.* To verify the conditions of Theorem 13 it is enough to check that $\mathrm{logit}(v) := \ln(v/(1-v))$ satisfies $\min(t, 1-t)^c \le \frac{\mathrm{d}}{\mathrm{d}t}\mathrm{logit}(t) \le C$. Since $\frac{\mathrm{d}}{\mathrm{d}t}\mathrm{logit}(t) = \frac{1}{t(1-t)}$, these conditions are satisfied with $c = 1$ and $C = 4$. $\square$

### B.3 GENERALIZED SMOOTHECE

We now generalize the definition of SmoothECE to other metrics, and show that it remains a consistent calibration measure with respect to its metric. Motivated by the logit example discussed above, a concrete way to introduce a non-trivial metric on a space of predictions $[0,1]$, is to consider a continuous and increasing function $h : [0,1] \to \mathbb{R} \cup \{\pm\infty\}$, and the metric obtained by pulling back the metric from $\mathbb{R}$ to $[0,1]$ through $h$, i.e. $d_h(u,v) := |h(u) - h(v)|$.

Using the isomorphism $h$ between $([0,1], d_h)$ and a subinterval of $(\mathbb{R}\cup\{\pm\infty\}, |\cdot|)$, we can introduce a generalization of the notion of $\mathsf{smECE}$, where the kernel-smoothing is being applied in the image of $h$.

More concretely, for a probability distribution $\mathcal{D}$ over $[0,1] \times \{0,1\}$, a kernel $K : \mathbb{R} \times \mathbb{R} \to \mathbb{R}_+$ and an increasing continuous map $h : [0,1] \to \mathbb{R} \cup \{\pm\infty\}$ we define

$$\hat{r}_{K,h}(t) := \frac{\mathbb{E}_{(f,y)} K(t, h(f))(f-y)}{\mathbb{E}_{(f,y)} K(t, h(f))}$$

$$\hat{\delta}_{K,h}(t) := \mathbb{E}_{(f,y)} K(t, h(f)).$$

Again, we define

$$\mathsf{smECE}_{K,h}(\mathcal{D}) := \int \hat{r}_{K,h}(t)\hat{\delta}_{K,h}(t)\,\mathrm{d}t,$$

which simplifies to

$$\mathsf{smECE}_{K,h}(\mathcal{D}) = \int \left| \mathbb{E}_{(f,y)\sim\mathcal{D}} K(t, h(f))(f-y) \right| \mathrm{d}t.$$

As it turns out, with the duality theorem in place (Theorem 12) the entire content of Section 3 can be carried over in this more general context without much trouble.

Specifically, if we define $\mathsf{smECE}_{\sigma,h} := \mathsf{smECE}_{K_{N,\sigma},h}$, where $K_{N,\sigma}$ is a Gaussian kernel with scale $\sigma$, then $\sigma \mapsto \mathsf{smECE}_{\sigma,h}(f,y)$ is non-increasing in $\sigma$, and therefore there is a unique fixed point $\sigma_*$ s.t. $\sigma_* = \mathsf{smECE}_{\sigma_*,h}(f,y)$.

We can now define $\mathsf{smECE}_{*,h}(f,y) := \sigma_*$, and we have the following generalization of Theorem 6, showing that SmoothECE remains a consistent calibration even under different metrics.

**Theorem 16.** *For any increasing and continuous function $h : [0,1] \to \mathbb{R} \cup \{\pm\infty\}$, if we define $d_h : [0,1]^2 \to \mathbb{R}_+$ to be the metric $d_h(u,v) = \max(|h(u) - h(v)|, 2)$ then*

$$\underline{\mathrm{dCE}}_{d_h}(\mathcal{D}) \lesssim \textsf{smECE}_{*,h}(\mathcal{D}) \lesssim \sqrt{\underline{\mathrm{dCE}}_{d_h}(\mathcal{D})}.$$

(Proof in Appendix D.7.)

Note that if the function $h$ is such that the associated metric $d_h$ satisfies the conditions of Theorem 13, as an additional corollary we can deduce that $\textsf{smECE}_{*,h}$ is also a consistent calibration measure in a standard sense.

### B.4 Obtaining perfectly calibrated predictor via post-processing

One of the appealing properties of the notion $\underline{\mathrm{dCE}}$ as it was introduced in Błasiok et al. (2023), was the theorem stating that if a predictor $(f,y)$ is close to calibrated, then in fact a nearby perfectly calibrated predictor can be obtained simply by post-processing all the predictions by a univariate function. Specifically, they showed that for a distribution $\mathcal{D}$ over $[0,1] \times \{0,1\}$, there is $\kappa : [0,1] \to [0,1]$ such that for $(f,y) \sim \mathcal{D}$ the pair $(\kappa(f), y)$ is perfectly calibrated and moreover $\mathbb{E}\,|\kappa(f) - f| \lesssim \sqrt{\underline{\mathrm{dCE}}(\mathcal{D})}$.

As it turns out, through the notion of $\textsf{smECE}_h$ we can prove a similar in spirit statement regarding the more general distances to calibration $\underline{\mathrm{dCE}}_{d_h}$. The only difference is that we allow the post-processing $\kappa$ to be a randomized function.

**Theorem 17.** *For any increasing function $h : [0,1] \to \mathbb{R} \cup \{\pm\infty\}$, and any distribution $\mathcal{D}$ supported on $[0,1] \times \{0,1\}$, there is a* probabilistic *function $\kappa : [0,1] \to [0,1]$ such that for $(f,y) \sim \mathcal{D}$, the pair $(\kappa(f), y)$ is perfectly calibrated and*

$$\mathbb{E}\, d_h(\kappa(f), f) \lesssim \textsf{smECE}_{*,h}(\mathcal{D}),$$

*where $d_h$ is the metric induced by $h$. In particular*

$$\mathbb{E}\, d_h(\kappa(f), f) \lesssim \sqrt{\underline{\mathrm{dCE}}_{d_h}(\mathcal{D})}.$$

*Proof.* Let us consider a distribution $\mathcal{D}$ over $[0,1] \times \{0,1\}$ and a monotone function $h$, such that $\textsf{smECE}_{h,*} = \sigma_*$.

First, let us define the randomized function $\kappa_1$: let $\pi_0 : \mathbb{R} \to \mathbb{R}$ be a projection of $\mathbb{R}$ to $h([0,1])$, and let $\eta \sim \mathcal{N}(0, \sigma_*)$. We define

$$\kappa_1(f) := h^{-1}(\pi_0(h(f) + \eta)).$$

We claim that this $\kappa_1$ satisfy the following two properties:

1. $\mathbb{E}_{(f,y) \sim \mathcal{D}} |d(f, \kappa'(f))| \lesssim \sigma_*$,

2. $\textsf{ECE}(\kappa'(f), y) \lesssim \sigma_*$.

Indeed, the first inequality is immediate:

$$\mathbb{E}[d(f, \kappa'(f)] = \mathbb{E}[|h(f) - \pi_0(h(f) + \eta)|] \leq \mathbb{E}\,|\eta| \leq \sigma_*.$$

The proof that $\textsf{ECE}(\kappa'(f), y) \lesssim \sigma_*$ is identical to the proof of Lemma 28, where such a statement was shown for the standard metric (corresponding to $h(x) = x$).

Finally, those two properties together imply the statement of the theorem: indeed, if $\textsf{ECE}(f', y) \leq \sigma_*$, we can take $\kappa_2(t) := \mathbb{E}[y|f' = t]$. In this case pair $(\kappa_2(f'), y)$ is perfectly calibrated, and by definition of $\textsf{ECE}$, we have $\mathbb{E}\,|\kappa_2(f') - f'| = \textsf{ECE}(f', y)$. Composing now $\kappa = \kappa_2 \circ \kappa_1$, we have

$$\mathop{\mathbb{E}}_{(f,y) \sim \mathcal{D}} |\kappa(f) - f| \leq \mathbb{E}[|\kappa_2 \circ \kappa_1(f) - \kappa_1(f)|] + \mathbb{E}[|\kappa_1(f) - f|] \lesssim \sigma_*.$$

Moreover distribution $\mathcal{D}'$ of $(\kappa(f), y)$ is perfectly calibrated.

$\square$

## C  Algorithms

In this section we discuss how smECE can be computed efficiently: for any $\sigma$, the quantity smECE$_\sigma$ can be approximated up to error $\varepsilon$ in time $\mathcal{O}(n + M^{-1} \log^{3/2} M^{-1})$ in the RAM model, where $M = \lceil \varepsilon^{-1} \sigma^{-1} \rceil$. In order to find an optimal scale $\sigma_*$, we need to perform a binary search, involving $\log \varepsilon^{-1}$ evaluations of smECE$_\sigma$.

The explicit procedure to compute smECE is given in Algorithm 2.

We shall first observe that the convolution with the reflected Gaussian kernel can be expressed in terms of a convolution with a shift-invariant kernel. This is useful, since such a convolution can be implemented in time $\mathcal{O}(M \log M)$ using Fast Fourier Transform, where $m$ is the size of the discretization.

**Claim 18.** *For any function $g : [0,1] \to \mathbb{R}$, the convolution with the reflected Gaussian kernel $g * \tilde{K}_{N,\sigma}$ can be equivalently computed as follows. Take an extension of $g$ to the entire real line $\tilde{g} : \mathbb{R} \to \mathbb{R}$ defined as $\tilde{g}(x) := g(\pi_R)(x))$. Then*

$$[g * \tilde{K}_{N,\sigma}](t) = [\tilde{g} * K_{N,\sigma}](t),$$

*where $K_{N,\sigma} : \mathbb{R} \times \mathbb{R} \to \mathbb{R}$ is the standard Gaussian kernel $K_\sigma(t_1, t_2) = \exp(-(t_1 - t_2)^2/2\sigma)/\sqrt{2\pi\sigma^2}$.*

*Proof.* Elementary calculation. $\qquad\square$

We can now restrict $\tilde{g}$ to the interval $[-T, T+1]$ where $T := \lceil \sqrt{\log(2\varepsilon^{-1})} \rceil$, convolve such a restricted $\tilde{g}$ with a Gaussian, and restrict the convolution in turn to the interval $\varepsilon$. Indeed, such a restriction introduces very small error, for every $t \in [0,1]$ we have.

$$[(\mathbf{1}_{[-T,T+1]} \cdot \tilde{g}) * K_{N,\sigma}](t) - [\tilde{g} * K_{N,\sigma}](t) \le (1 - \Phi(T/\sigma)) + (1 - \Phi((T+1)/\sigma))$$
$$\le \sqrt{2/\pi}(T/\sigma) \exp(-(T/\sigma)^2/2).$$

In practice, it is enough to reflect the function $g$ only twice, around two of the boundary points (corresponding to the choice $T = 1$). For instance, when $\sigma < 0.38$, the above bound implies that the introduced additive error is smaller than $\sigma^2$, and the error term rapidly improves as $\sigma$ is getting smaller.

Let us now discuss computation of smECE$_\sigma$ for a given scale $\sigma$. To this end, we discretize the interval $[0,1]$, splitting it into $M$ equal length sub-intervals. For a sequence of observations $(f_i, y_i)$ we round each $r_i$ to the nearest integer multiple of $1/M$, mapping it to a bucket $b_i = \text{round}(M f_i)$. In each bucket $b \in \{0, \ldots M\}$, we collect the residues of all observation falling in this bucket $h_b := \sum_{i:b_i=b} (f_i - y_i)$.

In the next step, we apply the Claim 18, and produce a wrapping $\tilde{h}$ of the sequence $h$ — extending it to integer multiples of $1/M$ in the interval $[-T, T+1]$ by pulling back $h$ through the map $\pi_R$.

The method smECE$_\sigma$ then proceeds to compute convolution $\tilde{h} * K$ with the discretization of the Gaussian kernel probability density function, i.e. $\tilde{K}_t := \exp(-t^2/2\sigma^2)$, and $K_t := \tilde{K}_t \left/ \sum_i \tilde{K}_t \right.$.

This convolution $\tilde{r} := h * K$ can be computed in time $\mathcal{O}(Q \log Q)$, where $Q = MT$, using a Fast Fourier Transform, and is implemented in standard mathematical libraries. Finally, we report the sum of absolute values of the residuals $\sum |\tilde{r}_i|$ as an approximation to smECE$_\sigma$ as an approximation to smECE$_\sigma$.

---

**Algorithm 1:** Efficient estimation of $\mathsf{smECE}_\sigma$, at fixed scale $\sigma$

---

**Function** Discretization($\{(f_i, z_i)\}_{i=1}^n$, $M$) **is**

    $h \leftarrow \text{zeros}(M + 1)$;

    **for** $i \in [n]$ **do**

        $b \leftarrow \text{round}(M f_i)$;

        $h_b \leftarrow h_b + z_i$;

    **end**

    **return** $h$;

**end**

**Function** Wrap($h$, $T$) **is**

    $M \leftarrow \text{len}(h)$;

    **for** $i \in [(2T + 1)M]$ **do**

        $j \leftarrow (i \mod 2M)$;

        **if** $j > M$ **then**

            $j \leftarrow 2M - j$;

        **end**

        $\tilde{h}_i \leftarrow h_j$;

    **end**

    **return** $\tilde{h}$;

**end**

**Function** $\mathsf{smECE}(\sigma, \{f_i, y_i\}_1^n)$ **is**

    $h \leftarrow \text{Discretization}(\{f_i, f_i - y_i\}, \lceil \sigma^{-1}\varepsilon^{-1} \rceil)$;

    $\tilde{h} \leftarrow \text{Wrap}(h, \lceil \sqrt{\log(2\varepsilon^{-1})} \rceil)$;

    $K \leftarrow \text{DiscreteGaussianKernel}(\sigma, \lceil \sigma^{-1}\varepsilon^{-1} \rceil)$;

    $\tilde{r} \leftarrow \tilde{h} * K$;

    **return** $\sum_{i=TM}^{(T+1)M-1} |\tilde{r}_i|$;

**end**

---

**Algorithm 2:** Efficient estimation of $\mathsf{smECE}_*$: using binary search over $\sigma$ to find a root of decreasing function $\mathsf{smECE}_\sigma - \sigma$.

---

**Data:** $(f_i, y_i)_1^n, \varepsilon$

**Result:** $\mathsf{smECE}_*(\{(f_i, y_i)\})$

$l \leftarrow 0$;

$u \leftarrow 1$;

**while** $u - l > \varepsilon$ **do**

    $\sigma \leftarrow (u + l)/2$;

    **if** $\mathsf{smECE}_\sigma(\{f_i, y_i\}) < \sigma$ **then**

        $u \leftarrow \sigma$;

    **else**

        $l \leftarrow \sigma$;

    **end**

**end**

**return** $u$;

---

## D   OMITTED PROOFS

### D.1   PROOF OF THEOREM 6

In this section we will prove Lemma 21 and Lemma 23, two main steps in the proof of Theorem 6, corresponding to respectively lower and upper bound. As it turns out, those two lemmas are true for a much wider class of kernels. The restriction on the kernel $K$ to be a Gaussian kernel stems from the monotonicity property (Lemma 28), which was convenient for us to define the scale invariant measure $\mathsf{smECE}_*$ by considering a fix-point scale $\sigma^*$. In Appendix D.2 we will show that the Reflected Gaussian kernel satisfies the conditions of Lemma 21 and Lemma 23.

We will first define a dual variant of dCE.

**Definition 19.** *We define the weak calibration error to be the maximal correlation of the residual* $(f - y)$ *with a 1-Lipschitz function and* $[-1, 1]$ *bounded function of a predictor, i.e.*

$$\mathrm{wCE}(\mathcal{D}) := \sup_{w \in \mathcal{L}} \mathbb{E}_{(f,y) \sim \mathcal{D}} w(f)(f - y),$$

*where* $\mathcal{L}$ *is a family of all 1-Lipschitz functions from* $[0, 1]$ *to* $[-1, 1]$.

To show that $\mathsf{smECE}^*$ is a consistent calibration measure we will heavily use the duality theorem proved in Błasiok et al. (2023) — the wCE and dCE are (up to a constant factor) equivalent. A similar statement is proved in this paper, in a greater generality (see Theorem 12).

**Theorem 20** (Błasiok et al. (2023)). *For any distribution* $\mathcal{D}$ *over* $[0, 1] \times \{0, 1\}$ *we have*

$$\underline{\mathrm{dCE}}(\mathcal{D}) \leq \mathrm{wCE}(\mathcal{D}) \leq 2\underline{\mathrm{dCE}}(\mathcal{D}).$$

Intuitively, this is useful since showing that a new measure $\mathsf{smECE}$ is a consistent calibration measure corresponds to upper and lower bounding it by polynomials of dCE. With the duality theorem above, we can use the minimization formulation dCE for one direction of the inequality, and the maximization formulation wCE for the other.

Indeed, we will first show that wCE is upper bounded by $\mathsf{smECE}$ if we add the penalty parameter for the "scale" of the kernel $K$.

**Lemma 21.** *Let* $U \subset \mathbb{R}$ *be (possible infinite) interval containing* $[0, 1]$ *and* $K : U \times U \to \mathbb{R}$ *be a non-negative symmetric kernel satisfying for every* $t_0 \in [0, 1]$, $\int K(t_0, t) \, \mathrm{d}t = 1$, *and* $\int |t - t_0| K(t, t_0) \, \mathrm{d}t \leq \gamma$. *Then*

$$\mathrm{wCE}(\mathcal{D}) \leq \mathsf{smECE}_K(\mathcal{D}) + \gamma.$$

*Proof.* Let us consider an arbitrary 1-Lipschitz function $w : [0, 1] \to [-1, 1]$, and take $\eta \sim K$ as in the lemma statement. Since kernel $K$ is nonnegative, and $\int K(t, t_0) \, \mathrm{d}t = 0$, we can sample triple $(\tilde{f}, f, y)$ s.t. $(f, y) \sim \mathcal{D}$, and $\tilde{f}$ is distributed according to density $K(\cdot, f)$. In particular $\mathbb{E}|\tilde{f} - f| \leq \gamma$.

We can bound now

$$\mathbb{E}_{(f,y) \sim \mathcal{D}}[w(f)(f - y)] \leq \mathbb{E}[w(\tilde{f})(f - y)] + \mathbb{E}|f - \tilde{f}||f - y|$$

$$\leq \gamma + \mathbb{E}\left[w(\tilde{f})(f - y)\right]. \tag{9}$$

We now observe that

$$\mathbb{E}[(f - y)|\tilde{f} = t] = \frac{\mathbb{E}_{f,y} K(t, f)(f - y)}{\mathbb{E}_{f,y} K(t, f)} = \hat{r}(t),$$

and the marginal density of $\tilde{f}$ is exactly

$$\mu_{\tilde{f}}(t) = \mathbb{E}_{(f,y) \sim \mathcal{D}} K(t, f) = \hat{\delta}(t).$$

This leads to

$$\mathbb{E}\left[w(\tilde{f})(f - y)\right] = \int w(t)\hat{r}(t)\hat{\delta}(t) \, \mathrm{d}t \leq \int |\hat{r}(t)|\hat{\delta}(t) \, \mathrm{d}t = \mathsf{smECE}_K(f, y). \tag{10}$$

Combining (11) and (12) we conclude the statement of this lemma.   $\square$

To show that $\mathsf{smECE}_K(\mathcal{D})$ is upper bounded by $\underline{\mathrm{dCE}}$, we will first show that $\mathsf{smECE}_K$ is zero for perfectly calibrated distributions, and then we will show that for well-behaved kernels $\mathsf{smECE}_K(\mathcal{D})$ is Lipschitz with respect the Wasserstein distance on the space of distributions.

**Claim 22.** *For any perfectly calibrated distribution $\mathcal{D}$ and for any kernel $K$ we have*

$$\mathsf{smECE}_K(\mathcal{D}) = 0.$$

*Proof.* Indeed, by the definition of $\hat{r}$ we have

$$\hat{r}(t) = \frac{\mathbb{E}_{f,y} K(f,t)(f-y)}{\mathbb{E}_{f,y} K(f,t)},$$

Since the distribution $\mathcal{D}$ is perfectly calibrated, we have $\mathbb{E}_{(f,y)\sim\mathcal{D}}[(f-y)|f] = 0$, hence

$$\mathbb{E}_{f,y}[K(f,t)(f-y)] = \mathbb{E}_f\left[\mathbb{E}_{(f,y)\sim\mathcal{D}}[K(f,t)(f-y)|f]\right] = \mathbb{E}_f\left[K(f,t)\mathbb{E}_{(f,y)\sim\mathcal{D}}[(f-y)|f]\right] = 0.$$

This means that the function $\hat{r}(t)$ is identically zero, and therefore

$$\mathsf{smECE}_K(\mathcal{D}) = \int_t |\hat{r}(t)||\hat{\delta}(t)|\,\mathrm{d}t = 0.$$

$\square$

**Lemma 23.** *Let $K$ be a symmetric, non-negative kernel, such that for and let $\lambda \leq 1$ be a constant such that for any $t_0, t_1 \in [0,1]$ we have $\int |K(t_0, t) - K(t_1, t)|\,\mathrm{d}t \leq |t_0 - t_1|/\lambda$. Let $\mathcal{D}_1, \mathcal{D}_2$ be a pair of distributions over $[0,1] \times \{0,1\}$. Then*

$$|\mathsf{smECE}_K(\mathcal{D}_1) - \mathsf{smECE}_K(\mathcal{D}_2)| \leq \left(\frac{1}{\lambda} + 1\right) W_1(\mathcal{D}_1, \mathcal{D}_2).$$

*Proof.* We have

$$\mathsf{smECE}_K(\mathcal{D}) = \int \left|\mathbb{E}_{(f,y)\sim\mathcal{D}}[K(t,f)(y-f)]\right|\,\mathrm{d}t.$$

If we have a coupling $(f_1, f_2, y_1, y_2)$ s.t. $\mathbb{E}[|f_1 - f_2| + |y_1 - y_2|] \leq \delta$, $(f_1, y_1) \sim \mathcal{D}_1$ and $(f_2, y_2) \sim \mathcal{D}_2$, then by triangle inequality we can decompose

$$|\mathsf{smECE}_K(\mathcal{D}_1) - \mathsf{smECE}_K(\mathcal{D}_2)| \leq \int \mathbb{E}_{(f_1,f_2,y_1,y_2)}[|K(t,f_1) - K(t,f_2)||y_1 - f_1|\,\mathrm{d}t$$

$$+ \int \mathbb{E}_{(f_1,f_2,y_1,y_2)}[K(t,f_2)(|f_1 - f_2| + |y_1 - y_2|]\,\mathrm{d}t.$$

We can bound those two terms separately

$$\int \mathbb{E}_{(f_1,f_2,y_1)}[|K(t,f_1)-K(t,f_2)||y_1-f_1|]\,\mathrm{d}t \leq \mathbb{E}_{(f_1,f_2,y_1)}\int |K(t,f_1)-K(t,f_2)|\,\mathrm{d}t \leq \frac{1}{\lambda}\mathbb{E}[|f_1-f_2|] \leq \delta/\lambda,$$

and similarly

$$\int \mathbb{E}[K(t,f_2)(|f_1-f_2|+|y_1-y_2|)]\,\mathrm{d}t = \mathbb{E}\left[\int_t K(t,f_2)\,\mathrm{d}t \cdot (|f_1-f_2|+|y_1-y_2|]\right] = \mathbb{E}[|f_1-f_2|+|y_1-y_2|] \leq \delta.$$

$\square$

**Corollary 24.** *Under the same assumptions on $K$ as in Lemma 23, for any distribution $\mathcal{D}$ over $[0,1] \times \{0,1\}$,*

$$\mathsf{smECE}_K(\mathcal{D}) \leq \left(\frac{1}{\lambda} + 1\right)\underline{\mathrm{dCE}}(\mathcal{D}).$$

*Proof.* By definition there is a perfectly calibrated distribution $\mathcal{D}'$, such that $W_1(\mathcal{D}, \mathcal{D}') \leq \underline{\mathrm{dCE}}(\mathcal{D})$. By Claim 22, $\mathsf{smECE}_K(\mathcal{D}') = 0$, and the corollary follows directly from Lemma 23. $\square$

### D.2 Facts about reflected Gaussian kernel

We wish to now argue that Lemma 21 and Lemma 23 imply the more specialized statements Lemma 7 and Lemma 8 respectively — the reflected Gaussian kernel $K_{N,\sigma}$ satisfies conditions of Lemma 21 and Lemma 23 with $\gamma$ and $\lambda$ proportional to $\sigma$. We

**Lemma 25.** *Reflected Gaussian kernel $\tilde{K}_{N,\sigma}$ defined by (2) satisfies*

1. *For every $t_0$, we have $\int \tilde{K}_{N,\sigma}(t, t_0)\,\mathrm{d}t = 1$.*

2. *For every $t_0$, we have $\int |t - t_0|\tilde{K}_{N,\sigma}(t, t_0)\,\mathrm{d}t \leq \sqrt{2/\pi}\sigma$.*

3. *For every $t_0, t_1$, we have $\int |\tilde{K}_{N,\sigma}(t, t_0) - \tilde{K}_{N,\sigma}(t, t_0)|\,\mathrm{d}t \leq |t_0 - t_1|/(2\sigma)$.*

*Proof.* For any given $t_0$, the function $\tilde{K}_{N,\sigma}(t_0, \cdot)$ is a probability density function of a random variable $\pi_R(t_0 + \eta)$ where $\eta \sim \mathcal{N}(0, \sigma)$ and $\pi_R : \mathbb{R} \to [0, 1]$ is defined in Section 1.1. In particular, we have $|\pi_R(x) - \pi_R(y)| \leq |x - y|$.

The property 1 is satisfied, since the $\tilde{K}_{N,\sigma}(\cdot, t_0)$ is a probability density function.

The property 2 follows since

$$\int |t - t_0|\tilde{K}_{N,\sigma}(t, t_0)\,\mathrm{d}t = \mathop{\mathbb{E}}_{\eta \sim \mathcal{N}(0,\sigma)} |\pi_R(t_0 + \eta) - t_0| = \mathop{\mathbb{E}}_{\eta \sim \mathcal{N}(0,\sigma)} |\pi_R(t_0 + \eta) - \pi_R(t_0)|$$
$$\leq \mathop{\mathbb{E}}_{\eta \sim \mathcal{N}(0,\sigma)} |\eta| = \sigma\sqrt{2/\pi}.$$

Finally, the property 2 again follows from the same fact for a Gaussian random variable: the integral $|\tilde{K}_{N,\sigma}(t, t_0) - \tilde{K}_{N,\sigma}(t, t_0)|$ is just a total variation distance between $\pi_R(t_0 + \eta)$ and $\pi_R(t_1 + \eta)$ where $\eta \sim \mathcal{N}(0, \sigma)$, but by data processing inequality we have

$$TV(\pi_R(t_0 + \eta), \pi_R(t_1 + \eta)) \leq TV(t_0 + \eta, t_1 + \eta) \leq |t_0 - t_1|/(2\sigma).$$

Where the last bound on the total variation distance between two one-dimension Gaussians is a special case of Theorem 1.3 in Devroye et al. (2018)[5]. $\square$

**Definition 26.** *We say that a paramterized family of kernels $K_\sigma : U \times U \to \mathbb{R}$ where $[0, 1] \subset U \subset \mathbb{R}$ is a proper kernel family if for any $\sigma_1 \leq \sigma_2$ there is a non-negative kernel $H_{\sigma_1, \sigma_2} : U \times U \to \mathbb{R}$, satisfying $\|H_{\sigma_1, \sigma_2}\|_{1 \to 1} \leq 1$ and $K_{\sigma_2} = K_{\sigma_1} * H_{\sigma_1, \sigma_2}$.*

*Here the notation $K * H$ is denotes*

$$[K * H](t_1, t_2) := \int_U K(t_1, t)H(t, t_2)\,\mathrm{d}t,$$

*and*

$$\|H\|_{1 \to 1} := \sup_{t_0 \in U} \int_U |H(t_0, t)|\,\mathrm{d}t.$$

**Claim 27.** *The family of reflected Gaussian kernels $\tilde{K}_{N,\sigma}$ is a proper kernel family, with*

$$\tilde{K}_{\sigma_1, N} = \tilde{K}_{\sigma_2, N} * \tilde{K}_{\sqrt{\sigma_1^2 - \sigma_2^2}, N}.$$

*Proof.* Let $\sigma_3 := \sqrt{\sigma_1^2 - \sigma_2^2}$, we wish to show that $\tilde{K}_{\sigma_1, N} = \tilde{K}_{\sigma_2, N} * \tilde{K}_{\sigma_3, N}$. In order to show this, it is enough to prove that for any $f$, we have $f * \tilde{K}_{\sigma_1, N} = f * \tilde{K}_{\sigma_2, N} * \tilde{K}_{\sigma_3, N}$. This is true by Claim 18, since this property holds for standard Gaussian kernel $K_{\sigma_2, N} * K_{\sigma_3, N} = K_{\sigma_1, N}$ (it is here equivalent to saying that for two independent random variables $Z_2 \sim \mathcal{N}(0, \sigma_2)$ and $Z_3 \sim \mathcal{N}(0, \sigma_3)$ we have $Z_2 + Z_2 \sim \mathcal{N}(0, \sigma_1)$). $\square$

---

[5]This special case, where the two variances are equal, is in fact an elementary calculation.

### D.3 USEFUL PROPERTIES OF SMECE.

**Lemma 28** (Monotonicity of smECE). *Let $K_\sigma$ be any proper kernel family parameterized by $\sigma$. If $\sigma_1 \leq \sigma_2$, then*

$$\mathsf{smECE}_{K_{\sigma_1}}(\mathcal{D}) \geq \mathsf{smECE}_{K_{\sigma_2}}(\mathcal{D}).$$

*Proof.* Let us define

$$h_\sigma(t) := \mathop{\mathbb{E}}_{(f,y) \sim \mathcal{D}} K_\sigma(t, f)(f - y) = \hat{r}(t)\hat{\delta}(t),$$

such that

$$\mathsf{smECE}_{K_\sigma}(\mathcal{D}) = \|h_\sigma\|_1 := \int |h_\sigma(t)| \, \mathrm{d}t.$$

Since $\sigma_1 \leq \sigma_2$ and $K_\sigma$ is a proper kernel family, we can write $K_{\sigma_2} = K_{\sigma_1} * H_{\sigma_1, \sigma_2}$.

We have now,

$$
\begin{aligned}
h_{\sigma_1} * H_{\sigma_1, \sigma_2} &= \left( \mathop{\mathbb{E}}_{(f,y)} (f - y) K_{\sigma_1}(\cdot, f) \right) * H_{\sigma_1, \sigma_2} \\
&= \mathop{\mathbb{E}}_{f,y} (f - y)[K_{\sigma_1} * H_{\sigma_1, \sigma_2}(\cdot, f)] = \mathop{\mathbb{E}}_{f - y} (f - y) K_{\sigma_2}(\cdot, f) \\
&= h_{\sigma_2}.
\end{aligned}
$$

On the other hand for any function $f$ we have $\|f * H_{\sigma_1, \sigma_2}\|_1 \leq \|f\|_1 \|H_{\sigma_1, \sigma_2}\|_{1 \to 1}$, and $\|H_{\sigma_1, \sigma_2}\|_{1 \to 1} \leq 1$ by the definition of proper kernel family. Therefore

$$\square$$

**Corollary 29.** *In particular for $\sigma_1 \leq \sigma_2$ we have $\mathsf{smECE}_{\sigma_2}(\mathcal{D}) \leq \mathsf{smECE}_{\sigma_1}(\mathcal{D})$.*

*Proof.* Reflected Gaussian kernels form a proper kernel family by Claim 27. $\square$

**Lemma 30.** *For any $\sigma$, we have $\widetilde{\mathsf{smECE}}_\sigma(\mathcal{D}) = \mathsf{smECE}_\sigma(\mathcal{D}) \pm \sigma\sqrt{2/\pi}$.*

*Proof.* Let

$$\hat{f}(t) := \frac{\mathbb{E}_{f,y} \, \tilde{K}_{N,\sigma}(t, f) f}{\mathbb{E}_{f,y} \, \tilde{K}_{N,\sigma}(t, f)}.$$

We have

$$
\begin{aligned}
|\widetilde{\mathsf{smECE}}_\sigma(f, y) - \mathsf{smECE}_\sigma(f, y)| &\leq \int |\hat{f}(t) - t|\hat{\delta}(t) \, \mathrm{d}t \\
&\leq \int \mathop{\mathbb{E}}_f [\tilde{K}_{N,\sigma}(t, f)|f - t|] \, \mathrm{d}t \\
&= \mathop{\mathbb{E}}_f \int K_\sigma(t, f)|f - t| \, \mathrm{d}t \\
&= \mathop{\mathbb{E}}_f \mathop{\mathbb{E}}_{Z \sim \mathcal{N}(f, \sigma)} |f - \pi_R(Z)| \\
&\leq \mathop{\mathbb{E}}_{Z \sim \mathcal{N}(0, \sigma)} |Z| = \sqrt{2/\pi}.
\end{aligned}
$$

$$\square$$

## D.4 EQUIVALENCE BETWEEN DEFINITIONS OF dCE FOR TRIVIAL METRICS

The $\underline{\text{dCE}}(\mathcal{D})$ was defined in Błasiok et al. (2023) as a Wasserstein distance to the set of perfectly calibrated distributions over $X := [0,1] \times \{0,1\}$, where $X$ is equipped with a metric

$$d_1((f_1, y_1), (f_2, y_2)) := \begin{cases} |f_1 - f_2| & \text{if } y_1 = y_2 \\ \infty & \text{otherwise} \end{cases}.$$

While generalizing the notion to that of $\underline{\text{dCE}}_d$, where $d$ is a general metric on $[0,1]$, we chose a different metric on $X$ (specifically, we put a different metric on the second coordinate), that is $\tilde{d}((f_1, y_1), (f_2, y_2)) = d(f_1, f_2) + |y_1 - y_2|$.

As it turns out, for the case of a trivial metric on the space of predictions, this choice is inconsequential, but the new definition has better generalization properties.

**Claim 31.** *For the metric $\ell_1(f_1, f_2) = |f_1 - f_2|$, we have $\underline{\text{dCE}}(\mathcal{D}) \lesssim \underline{\text{dCE}}_{\ell_1}(\mathcal{D}) \leq \underline{\text{dCE}}(\mathcal{D})$, for some universal constant c.*

*Proof.* The lower bound $\underline{\text{dCE}}_{\ell_1} \leq \underline{\text{dCE}}$ is immediate, since $\underline{\text{dCE}}_{\ell_1}$ is a distance of $\mathcal{D}$ to $\mathcal{P}$ with respect to a Wasserstein distance induced by the metric $d_1$ on $[0,1] \times \{0,1\}$, $\underline{\text{dCE}}$ is the Wasserstein distance with respect to the metric $d_2$, and we have a pointwise bound $d_1(u,v) \leq d_2(u,v)$, implying $W_{1,d_1}(\mathcal{D}_1, \mathcal{D}_2) \leq W_{1,d_2}(\mathcal{D}_1, \mathcal{D}_2)$.

The other bound follows from Theorem 12 and Theorem 20 — $\underline{\text{dCE}}$ and $\underline{\text{dCE}}_{\ell_1}$ are within constant factor from $\text{wCE}_{\ell_1}$. $\qquad\square$

## D.5 PROOF OF LEMMA 14

*Proof.* Let us take $w(x) : [0,1] \to [-1,1]$ as in the definition of $\text{wCE}_d$, a 1-Lipschitz function with respect to the metric $d$, such that $\mathbb{E}(y - f)w(f) = \text{wCE}_d(f, y) = \varepsilon$.

We wish to show that $\text{wCE}(f, y) \gtrsim \varepsilon^{c+1}$. Indeed, let us take $\tilde{w}(X) := w(\pi_I(x))$ where $I := [\gamma, 1 - \gamma]$, $\pi_I : [0,1] \to I$ is a projection onto the interval $I$, and $\gamma := \varepsilon/C$ for some large constant $C$.

Note that $\tilde{w}$ is $\mathcal{O}(\varepsilon^{-c})$-Lipschitz with respect to the standard metric on $[0,1]$. If $\mathbb{E}(f - y)\tilde{w}(f) \geq \varepsilon/2$, we immediately have $\text{wCE}(f, y) \gtrsim \varepsilon^{c+1}$ (we can use $\tilde{w}/L$ as a test function, where $L = \mathcal{O}(\varepsilon^{-c})$ is a Lipcshitz constant for function $\tilde{w}$). Otherwise $\mathbb{E}(f - y)(w(f) - \tilde{w}(f)) \geq \varepsilon/2$. Let us call $w_2 := (w - \tilde{w})/2$, such that $\mathbb{E}(f - y)w_2(f) \geq \varepsilon/4$, and moreover $\text{supp}(w_2) \subset [0,1] \setminus I$, where $w_2$ is 1-Lipschitz with respect to $d$.

Since $[0,1] \setminus I$ has two connected components $[0, \gamma)$ and $(1 - \gamma, 1]$, on one of those two connected components correlation between the residual $(y - f)$ and $w_2$ has to be at least $\varepsilon/8$. Since the other case is analogous, let us assume for concreteness, that

$$\mathbb{E}(y - f)w_3(f) \geq \varepsilon/8,$$

where $w_3(x) = w_2(x)$ for $x \in [0, \gamma)$ and $w_3(x) = 0$ otherwise.

We will show that this implies $\Pr(f \leq \gamma \wedge y = 1) \gtrsim \varepsilon$, and refer to Claim 32 to finish the argument.

Indeed

$$\mathbb{E}(y - f)w_3(f) \leq \mathbb{E}\left[(1 - f)\mathbf{1}[f \leq \gamma \wedge y = 1]\right] + \mathbb{E}\left[f\mathbf{1}[f \leq \gamma \wedge y = 0]\right] \leq \Pr(f \leq \gamma \wedge y = 1) + \gamma,$$

hence

$$\Pr(f \leq \gamma \wedge y = 1) \geq \varepsilon/8 - \gamma \geq \varepsilon/16,$$

where we finally specify $\gamma := \varepsilon/32$.

To finish the proof, it is enough to show the following

**Claim 32.** *For a random pair $(f, y)$ of prediction and outcome, if $\Pr(f \leq \gamma \wedge y = 1) \geq \varepsilon$ or $\Pr(f \geq 1 - \gamma \wedge y = 0) \geq \varepsilon$, where $\gamma = \varepsilon/8$, then $\text{wCE}(f, y) \gtrsim \varepsilon^2$.*

*Proof.* We will only consider the case $\Pr(f \leq \gamma \wedge y = 1) \geq \varepsilon$. The other case is identical.

Let us take $w(x) := \max(1 - x/2\gamma, 0)$. We have

$$\mathbb{E}(y - f)w(f) \geq \frac{1}{2}\Pr(f \leq \gamma \wedge y = 1) - 2\gamma \Pr(f \leq \gamma \wedge y = 0) \geq \varepsilon/2 - 2\gamma \geq \varepsilon/4.$$

Since $w$ is $\mathcal{O}(1/\varepsilon)$-Lipschitz, we have $\mathrm{wCE}(f, y) \gtrsim \varepsilon^2$. $\qquad\qquad\square$

$\hfill\square$

## D.6 SAMPLE COMPLEXITY — PROOF OF THEOREM 9

**Lemma 33.** *Let $X : [0, 1] \to \mathbb{R}$ be a random function, satisfying with probability 1, $\|X\|_1 := \int_0^1 |X(t)| \, \mathrm{d}t \leq 1$ and $\sup_t X(t) \leq \sigma$. Assume moreover that for every $t$, we have $\mathbb{E}[X(t)] = 0$.*

*Consider now $m$ independent realizations $X_1, X_2, \ldots X_m : [0, 1] \to \mathbb{R}$, each identically distributed as $X(t)$, and finally let*

$$\overline{X}(t) := \frac{1}{m}\sum X_i(t).$$

*Then*

$$\mathbb{E}\left[\|\overline{X}(t)\|_1^2\right] \leq \frac{1}{\sigma m}.$$

*Proof.* By Cauchy-Schwartz inequality $\|X\|_1 \leq \|X\|_2\|\mathbf{1}\|_2 = \|X\|_2$, hence

$$\begin{aligned}
\mathbb{E}[\|\overline{X}\|_1^2] &\leq \mathbb{E}[\|\overline{X}\|_2^2] = \mathbb{E}\left[\int \overline{X}(t)^2 \, \mathrm{d}t\right] \\
&= \int \mathbb{E}[\overline{X}(t)^2] \, \mathrm{d}t \\
&= \frac{1}{m}\int \mathbb{E}[X(t)^2] \, \mathrm{d}t \\
&= \frac{1}{m}\mathbb{E}[\|X\|_2^2] \leq \frac{1}{m}\mathbb{E}[\|X\|_1\|X\|_\infty] \leq \frac{1}{\sigma m}.
\end{aligned}$$

$\hfill\square$

*Proof of Theorem 9.* Let us first focus on the case $\sigma = \sigma_0$. For a pair $(f, y) \in [0, 1] \times \{0, 1\}$, let us define $X_{f,y}^{(\sigma_0)} : [0, 1] \to \mathbb{R}$ as

$$X_{f,y}^{(\sigma_0)}(t) := \tilde{K}_{\sigma_0}(f, t)(f - y).$$

Note that $\mathsf{smECE}_{\sigma_0}(\hat{\mathcal{D}}) = \|\sum_i X_{f_i,y_i}^{(\sigma_0)}/m\|_1$, and similarly $\mathsf{smECE}(\mathcal{D}) = \|\mathbb{E}_{f,y\sim\mathcal{D}} X_{f,y}^{(\sigma_0)}\|_1$.

Define $\tilde{X}_i^{(\sigma_0)} := X_{f_i,y_i}^{(\sigma_0)} - \mathbb{E}_{f,y\sim\mathcal{D}} X_{f,y}^{(\sigma_0)}$ — this is a random function, since $(f_i, y_i)$ is chosen at random from distribution $\mathcal{D}$, and note that:

1. Random functions $\tilde{X}_i^{(\sigma_0)}$ for $i \in \{1, \ldots, m\}$ are independent and identically distributed.

2. With probability 1, we have $\|\tilde{X}_i^{(\sigma_0 0)}\|_1 \leq 2\max_f \|\tilde{K}_{\sigma_0}(f, \cdot)\|_1 = 2$.

3. Similarly, with probability 1 we have $\|\tilde{X}_i^{(\sigma_0)}\|_\infty \leq 2\sup_{t_1,t_2} \tilde{K}_{\sigma_0}(t_1, t_2) \leq 2\sigma_0^{-1}$.

4. For any $t \in [0, 1]$ and $i \in \{1, \ldots, m\}$, we have $\mathbb{E}[\tilde{X}_i^{(\sigma_0)}(t)] = 0$.

Therefore, we can apply Lemma 33 to deduce

$$\mathbb{E}\left[\left\|\frac{1}{m}\sum\tilde{X}_i\right\|_1^2\right] \le \frac{1}{\sigma_0 m},$$

hence, if $m \gtrsim \varepsilon^{-2}\sigma_0^{-1}$, by Chebyshev inequality with probability at least $2/3$ we can bound $\|\sum_i \tilde{X}_i^{(\sigma_0)}/m\|_1 \le \varepsilon$, and if this event holds, using triangle inequality

$$\mathsf{smECE}_{\sigma_0}(\mathcal{D}) - \mathsf{smECE}_{\sigma_0}(\hat{\mathcal{D}})| \le \|\sum_i \tilde{X}_i^{(\sigma_0)}\|/m\varepsilon.$$

Finally, for $\sigma > \sigma_0$, note that $X_i^{(\sigma)} = X_i^{(\sigma_0)} * \tilde{K}_{N,\sqrt{\sigma^2-\sigma_0^2}}$ (Claim 27) and therefore as soon as $\|\sum \tilde{X}_i^{(\sigma_0)}\| \le \varepsilon$, we also have

$$\|\sum_i \tilde{X}_i^{(\sigma)}/m\|_1 = \|\sum_i \tilde{X}_i^{(\sigma_0)} * \tilde{K}_{N,\sqrt{\sigma^2-\sigma_0^2}}\|_1$$

$$\le \|\sum_i \tilde{X}_i^{(\sigma_0)}\|_1 \|\tilde{K}\|_{1\to1} \le \varepsilon,$$

where

$$\|\tilde{K}\|_{1\to1} := \sup_{t_1} \int_{t_2} |\tilde{K}(t_1,t_2)|\,\mathrm{d}t \le 1.$$

This implies $|\mathsf{smECE}_\sigma(\mathcal{D}) - \mathsf{smECE}_\sigma(\hat{\mathcal{D}})| < \varepsilon$ for all $\sigma \ge \sigma_0$.

Finally, if $\mathsf{smECE}_*(\mathcal{D}) = \sigma_* \ge \sigma_0$, we have $\mathsf{smECE}_{\sigma_*}(\mathcal{D}) = \sigma_*$, hence $\mathsf{smECE}_{\sigma_*}(\hat{\mathcal{D}}) \ge \sigma_* - \varepsilon$, and by monotonicity $\mathsf{smECE}_{\sigma_*-\varepsilon}(\hat{\mathcal{D}}) \ge \sigma_* - \varepsilon$, implying $\mathsf{smECE}_*(\hat{\mathcal{D}}) \ge \sigma_* - \varepsilon$. Identical argument shows $\mathsf{smECE}_*(\hat{\mathcal{D}}) \le \sigma_* + \varepsilon$.

$\square$

### D.7 Proof of Theorem 16

The Lemma 21 and Lemma 23 have their correspondent versions in the more general setting where a metric is induced on the space of predictions $[0,1]$ by a monotone function $h : [0,1] \to \mathbb{R}$ — the proofs are almost identical to those supplied in the special case, except we need to use the more general version of the duality theorem between wCE and dCE, with respect to a metric $d$ (Theorem 12).

**Lemma 34.** *Let $h$ be an increasing function $h : [0,1] \to \mathbb{R} \cup \{\pm\infty\}$ and $d_h(u,v) = |h(u) - h(v)|$ be the induced metric on $[0,1]$. Assume moreover that $K(t)$ is a probability density function, such that for $\eta \sim K$ we have $\mathbb{E}|\eta| = \gamma$. Finally, let us abuse the notation to define the associated kernel on $\mathbb{R} \times \mathbb{R}$ as $K(x,y) := K(x-y)$. Then*

$$\mathrm{wCE}_d(f,y) \le \mathsf{smECE}_{K,d_h}(f,y) + \gamma.$$

*Proof.* Let us consider an arbitrary 1-Lipschitz function $w : [0,1] \to [-1,1]$ with respect to $d_h$, and take $\eta \sim K$ as in the lemma statement. Let $I = h([0,1]$, and let us take $\tilde{w} : I \to \mathbb{R}$ to be given by $\tilde{w}(t) = w(h^{-1}(t))$, and note that $\tilde{w}$ is Lipschitz, since $w$ was Lipschitz with respect to $d_h$. We can therefore extend $\tilde{w}$ to a $[-1,1]$ valued Lipschitz function on the entire line $\mathbb{R}$. Now for any $f, \eta$ we have $|\tilde{w}(h(f)+\eta) - \tilde{w}(h(f))| \le |\eta|$, and we can bound

$$\mathbb{E}_{(f,y)\sim\mathcal{D}}[w(f)(f-y)] \le \mathbb{E}[\tilde{w}(h(f)+\eta)(f-y)] + \mathbb{E}|\eta||f-y|$$

$$\le \mathbb{E}|\eta| + \mathbb{E}\left[\mathbb{E}[\tilde{w}(h(f)+\eta)(f-y)]\right]$$
$$\le \gamma + \mathbb{E}\left[\mathbb{E}[\tilde{w}(h(f)+\eta)(f-y)]\right]. \quad (11)$$

We now observe that

$$\mathbb{E}[(f-y)|h(f)+\eta = t] = \frac{\mathbb{E}_{f,y}\,K(t,h(f))(f-y)}{\mathbb{E}_{f,y}\,K(t,h(f))} = \hat{r}(t),$$

and

$$\mu_{f+\eta}(t) = \underset{f,y}{\mathbb{E}}\, K(t, h(f)) = \hat{\delta}(t),$$

where $\mu_{f+\eta}$ is the measure of $h(f) + \eta$.

This leads to

$$\mathbb{E}\left[\tilde{w}(h(f) + \eta)(f - y)\right] = \int \tilde{w}(t)\hat{r}(t)\hat{\delta}(t)\,\mathrm{d}t \leq \int |\hat{r}(t)|\hat{\delta}(t)\,\mathrm{d}t = \mathsf{smECE}_{K,d_h}(f, y). \quad (12)$$

Combining (11) and (12) we conclude the statement of this lemma. $\qquad\square$

**Lemma 35.** *Let $h$ be an increasing function $h : [0, 1] \to \mathbb{R} \cup \{\pm\infty\}$, and $d_h(u, v) := |h(u) - h(v)|$ be the induced metric on $[0, 1]$.*

*Let $K(t)$ be a probability density function of a random variable, such that for $\eta \sim K$ we have $\mathbb{E}\,\eta = 0$, and let $\lambda \leq 1$ be a constant such that for any $\varepsilon$, we have $\mathrm{TV}(\eta, \eta + \varepsilon) \leq \varepsilon/\lambda$, finally by the abuse of notation let us define te assocated kernel on $\mathbb{R} \times \mathbb{R}$ as $K(x, y) := K(x - y)$. Then Then*

$$\mathsf{smECE}_{K,h}(\mathcal{D}) \leq \left(\frac{1}{\lambda} + 1\right)\underline{\mathrm{dCE}}_{d_h}(\mathcal{D}).$$

### D.8 GENERAL DUALITY THEOREM (PROOF OF THEOREM 12)

Let $\mathcal{P} \subset \Delta([0, 1] \times \{0, 1\})$ be the family of perfectly calibrated distributions. This set is cut from the full probability simplex $\Delta([0, 1] \times \{0, 1\})$ by a family of linear constraints, specifically $\mu \in \mathcal{P}$ if and only if

$$\forall t, (1 - t)\mu(t, 1) - t\mu(t, 0) = 0.$$

**Definition 36.** *Let $\mathcal{F}(H, \mathbb{R})$ be a family of all functions from $H$ to $\mathbb{R}$. For a convex set of probability distributions $\mathcal{Q} \subset \Delta(H)$, we define $\mathcal{Q}^* \subset \mathcal{F}(H, \mathbb{R})$ to be a set of all functions $q$, s.t. for all $\mathcal{D} \in \mathcal{Q}$ we have $\mathbb{E}_{x\sim\mathcal{D}}\,q(x) \leq 0$.*

**Claim 37.** *The set $\mathcal{P}^* \subset \mathcal{F}([0, 1] \times \{0, 1\}, \mathbb{R})$ is given by the following inequalities. A function $H \in \mathcal{P}^*$ if and only if*

$$\forall t, \underset{y\sim\mathrm{Ber}(t)}{\mathbb{E}}\, H(t, y) \leq 0.$$

$\qquad\square$

**Lemma 38.** *Let $W_1(\mathcal{D}_1, \mathcal{D}_2)$ be the Wasserstein distance between two distributions $\mathcal{D}_1, \mathcal{D}_2 \in \Delta([0, 1] \times, \{0, 1\})$ with arbitrary metric $d$ on the set $[0, 1] \times \{0, 1\}$, and let $Q \subset \Delta([0, 1] \times \{0, 1\})$ be a convex set of probability distributions.*

*The value of the minimization problem*

$$\min_{\mathcal{D}_1 \in \mathcal{Q}} W_1(\mathcal{D}_1, \mathcal{D})$$

*is equal to*

$$\max \underset{(f,y)\sim\mathcal{D}}{\mathbb{E}}\, H(f, y)$$
$$s.t. \quad H \text{ is Lipschitz with respect to } d,$$
$$H \in \mathcal{Q}^*.$$

*Proof.* Let us consider a linear space $\mathcal{V}$ of all finite signed Radon measures on $X := [0, 1] \times \{0, 1\}$, satisfying $\mu(X) = 0$. We equip this space with the norm $\|\mu\|_{\mathcal{V}} := \mathrm{EMD}(\mu_+, \mu_-)$ for measures s.t. $\mu_+(X) = 1$ (and extended by $\|\lambda\mu\|_{\mathcal{V}} = \lambda\|\mu\|_{\mathcal{V}}$ to entire space). The dual of this space is $\mathrm{Lip}_0(X)$ — space of all Lipschitz functions on $X$ which are 0 on some fixed base point $x_0 \in X$ (the choice of base point is inconsequential). The norm on $\mathrm{Lip}_0(X)$ is $\|W\|_L$ given by the Lipschitz constant of $W$ (see Chapter 3 in Weaver (2018) for proofs and more extended discussion).

For a function $H$ on $X$ and a measure $\mu$ on $X$, we will write $H(\mu)$ to denote $\int W\,\mathrm{d}\mu$.

The weak duality is clear: for any Lipschitz function $H \in \mathcal{Q}^*$, and any distribution $\mathcal{D}_1 \in \mathcal{Q}$ we have $H(\mathcal{D}) \leq H(\mathcal{D}_1) + W_1(\mathcal{D}_1, \mathcal{D}) = W_1(\mathcal{D}_1, \mathcal{D})$.

For the strong duality, we shall now apply the following simple corollary of Hahn-Banach theorem.

**Claim 39** (Deutsch & Maserick (1967), Theorem 2.5)**.** *Let $(X, \| \cdot \|_X)$ be a normed linear space, $x_0 \in X$, and $P \subset X$ a convex set, and let $d(x, P) := \inf_{p \in P} \|x - p\|_X$. Then there is $w \in X^*$, such that $\|w\|_{X^*} = 1$ and $\inf_{p \in P} w(p) - w(x) = d(x, P)$.*

Take a convex set $P \subset \mathcal{V}$ given by $P := \{\mathcal{D} - q : q \in \mathcal{Q}\}$. Clearly $\min_{D_1 \in \mathcal{Q}} W_1(\mathcal{D}, \mathcal{D}_1) = d(0, P)$ by definition of the space $\mathcal{V}$, and hence using the claim above, we deduce

$$d(0, P) = \max_{\tilde{H} \in \mathrm{Lip}_0 : \|\tilde{H}\|_L = 1} \inf_{p \in P} \tilde{H}(p).$$

Taking $\tilde{H}$ which realizes this maximum, we can now consider a shift $H := \tilde{H} - \sup_{q \in \mathcal{Q}} \tilde{H}(Q)$, so that $H \in \mathcal{Q}^*$, and verify

$$\min_{D_1 \in \mathcal{Q}} W_1(\mathcal{D}, \mathcal{D}_1) = d(0, P) = \inf_{p \in P} \tilde{H}(p) = \tilde{H}(\mathcal{D}) - \sup_{q \in \mathcal{Q}} \tilde{H}(q) = H(\mathcal{D}).$$

$\square$

**Corollary 40.** *For any metric $d$ on $[0, 1] \times \{0, 1\}$, the $\underline{\mathrm{dCE}}_d(\mathcal{D})$ is equal to the value of the following maximization program*

$$\max \quad \mathbb{E}_{(f,y) \sim \mathcal{D}} H(f, y)$$
$$\text{s.t.} \quad H \text{ is Lipschitz with respect to } d$$
$$\forall t, \quad \mathbb{E}_{y \sim \mathrm{Ber}(t)} H(t, y) \leq 0.$$

**Lemma 41.** *For any metric $d$ on $[0, 1]$ if we define $\hat{d}$ to be a metric on $[0, 1] \times \{0, 1\}$ given by $\hat{d}((f_1, y_1), (f_2, y_2)) := d(f_1, f_2) + |y_1 - y_2|$, we have*

$$\mathrm{wCE}_d(\mathcal{D}) \geq \underline{\mathrm{dCE}}_{\hat{d}}(\mathcal{D})/2$$

*Proof.* We shall compare the value of $\mathrm{wCE}_d(\mathcal{D})$ with the optimal value of the dual as in Corollary 40.

Let us assume that for a distribution $\mathcal{D}$ we have a function $H : [0, 1] \times \{0, 1\} \to \mathbb{R}$, s.t. $\mathbb{E}_{(f,y) \sim \mathcal{D}} H(f, y) = \mathrm{OPT}$, which is Lipschitz with respect to $\hat{d}$. We wish to find a function $w : [0, 1] \to [-1, 1]$ which is Lipschitz with respect to $d$, s.t.

$$\mathbb{E}_{f,y} (f - y) w(f) \geq \mathrm{OPT}/2.$$

Let us take

$$w(f) := H(f, 0) - H(f, 1).$$

We will show instead that $w$ is 2-Lipschitz, $[-1, 1]$ bounded and satisfies $\mathbb{E}_{f,y}(f - y)w(f) \geq \mathrm{OPT}$, and the statement of the lemma will follow by scaling.

Let us define $w(f) := H(f, 0) - H(f, 1)$. The condition

$$\forall f, \quad \mathbb{E}_{y \sim \mathrm{Ber}(f)} H(f, y) \leq 0$$

is equivalent to $f w(f) \geq H(f, 0)$. Hence

$$H(f, y) = y H(f, 1) + (1 - y) H(f, 0) = H(f, 0) - y w(f) \leq (f - y) w(f),$$

which implies $\mathbb{E}(f - y) w(f) \geq \mathbb{E} H(f, y)$.

Moreover, the function $w(f)$ is bounded by construction of the metric $\hat{d}$ and the assumption that $H(f, y)$ was Lipschitz. Indeed $|w(f)| = |H(f, 0) - H(f, 1)| \leq \hat{d}((f, 0), (f, 1)) \leq 1$. $\square$

## E    CONSISTENT CALIBRATION MEASURES AND SMOOTHECE

Here we elaborate on the shortcomings of ECE, and resolution offered by SmoothECE, and the notion of a *consistent calibration measure* (Błasiok et al., 2023). One fundamental issue with the ECE is: it is *discontinuous* in the underlying predictor, so a small change in the predictor can cause a large change in its ECE. A simple example of this phenomenon was presented in the Introduction of Błasiok et al. (2023) (also formalized as Lemma 4.8 in the same work). Here, we will demonstrate this discontinuity visually, with a related example.

In Figure 3, we construct three different distributions that are each small perturbations of one another. The top row shows samples $(f_i, y_i) \in [0, 1] \times \{0, 1\}$ from each distribution. The first distribution (first column) is nearly perfectly calibrated. This is is evident from both the smoothECE diagram (middle row) and the binnedECE diagram (bottom row). To construct the second and third distributions, we shift each positive sample $(y = 1)$ slightly to the right, and each negative sample $(y = 0)$ slightly to the left. No sample moves more than $0.05$ from its initial position in Distribution 1, so this is a small perturbation to the predictions $f_i$. We see that the smooth reliability diagram in the middle row does not change much between all three distributions, and the smECE metric itself also stays nearly constant (the smECE metric is listed in the upper-left of each plot).

However, by Distribution 3, the binned reliability diagram (bottom row) has changed drastically, and appears to be significantly mis-calibrated. Moreover, the binned ECE metric (with 20 bins) has changed from $0.06$ in the first distribution to $0.31$ in the third distribution. The binned ECE thus incorrectly reports that Distribution 3 is far from calibrated. This essentially occurs because we have shifted the positive and negative samples into disjoint bins by Distribution 3.

This example demonstrates how standard (binned) ECE is *not robust* to small perturbations in the predictor, and can severely over-estimate the calibration error. Smooth ECE, on the other hand, does not suffer this flaw. The formal definition of a *consistent calibration measure* in Błasiok et al. (2023) enforces this robustness/continuity property, in addition to other natural theoretical properties. We refer to the exposition in Błasiok et al. (2023) for further theoretical details.

### E.1    STATISTICAL CONSISTENCY

Our estimator of SmoothECE also satisfies the classical criteria of *statistical consistency*. Specifically Theorem 9 shows that as the number of samples $m \to \infty$, the finite-sample estimation error $\varepsilon \to 0$. In fact, Theorem 9 is stronger than just asymptotic consistency— it provides a quantitative generalization bound.

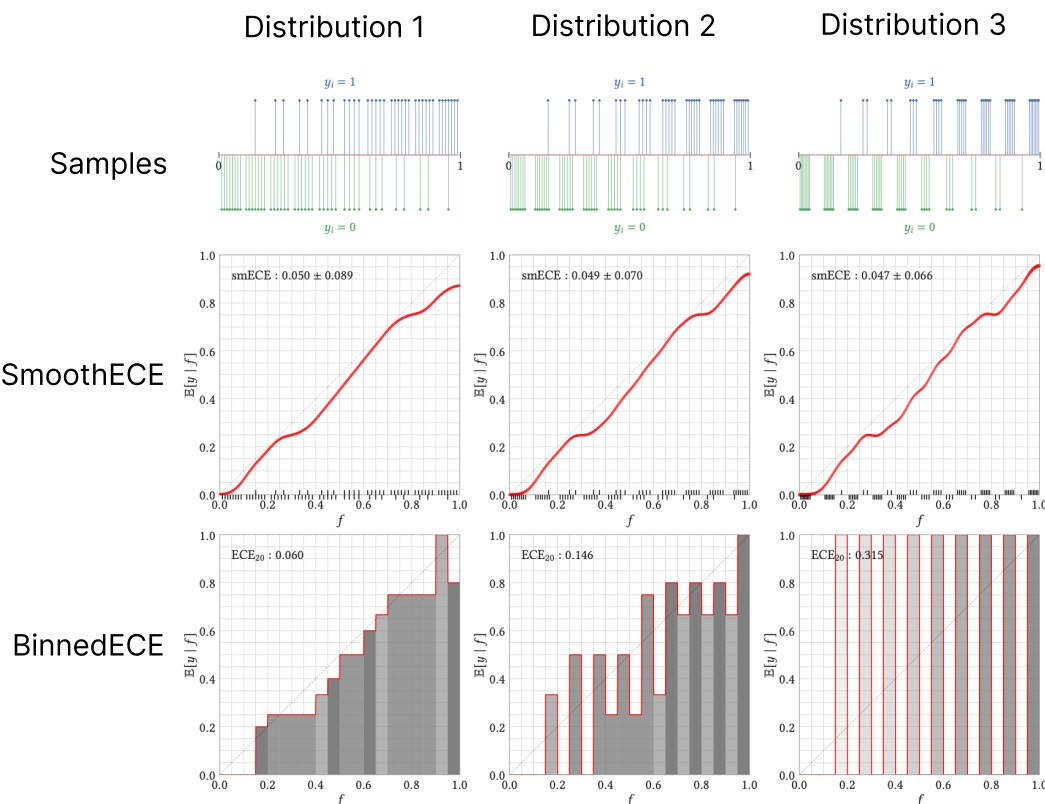

Figure 3: **Continuity Demonstration.** A demonstration of the discontinuous behavior of standard (binned) ECE, and how this is addressed by our Smooth ECE. We construct three different distributions that are each small perturbations of one another. The top row shows samples $(f_i, y_i) \in [0, 1] \times \{0, 1\}$ from each distribution. The first distribution is nearly perfectly calibrated. To construct the second and third distributions, we shift each positive sample $(y = 1)$ slightly to the right, and each negative sample $(y = 0)$ slightly to the left. No sample moves more than $0.05$ from its initial position in Distribution 1, so this is a small perturbation to the predictions $f_i$. We see that the smooth reliability diagram in the middle row does not change much between all three distributions. However, the binned ECE in the bottom row changes drastically, and reports severe miscalibration on Distribution 3 — even though all three distributions are actually close to calibrated. **See Section E for more details.**

