# OpenReview forum: "Smooth ECE: Principled Reliability Diagrams via Kernel Smoothing"
_ICLR.cc/2024/Conference — ICLR 2024 poster_

### Official Review · Reviewer_MdfL · 2023-10-31

**Soundness:** 3 good
**Presentation:** 2 fair
**Contribution:** 3 good
**Rating:** 6
**Confidence:** 3

**Summary:**

The paper studies the limitations of well-known expected calibration error (ECE) and the associated reliability diagram used for quantifying the miscalibration of a machine learning model. Among the core limitations of ECE, the fundamental ones are: discontinuous functional and impossible to estimate efficiently from samples. The paper proposes smoothECE and a regression method, which is ECE of the smoothed version of the original distribution and so the associated reliability diagram reflects a smoothed estimate of the calibration function. It claims that kernel smoothing allows realizing a consistent calibration measure and it provides a way of choosing the kernel bandwidth. The paper shows that SmoothECE has desirable mathematical properties, including consistent calibration measure and being sample and runtime efficient. Experiments have been performed on Deep Networks, Solar Flares, meteorological data, and synthetic data to showcase the comparison between ECE and proposed SmoothECE.

**Strengths:**

- Calibration is an important aspect of probabilistic predictors as they allow building a certain level of trust in the model’s predictions, therefore studying the limitations of prevalent miscalibration measures i.e. ECE is an important and relevant research direction.

- The idea of applying kernel smoothing with automatic selection kernel bandwidth to the original distribution is simple. It has been shown that the resulting smoothECE has desirable mathematical properties.

- Experimental results have been shown on different scenarios, including deep networks, rain forecasts, and solar flares forecasts.

**Weaknesses:**

- It would be interesting to see the results of SmoothECE on imbalanced (image) datasets such as SVHN.

- The overall contributions of the paper seem a bit limited as it boils down to applying (Gaussian) kernel to the original distribution for obtaining a smoothed estimate of it (which is already done in the prior work) and a technique to determine the kernel bandwidth.

- The empirical results are missing experiments on more image datasets, especially that cover out-of-distribution scenarios, and different types of networks.

**Questions:**

- It is a bit difficult to understand that, how the smooth reliability is visually more interpretable than the binned reliability.

- To what extent, this smoothECE can be trusted as a standalone miscalibration measure?

**Details Of Ethics Concerns:**

No ethics concerns.

---

> ### Author Response · Authors · 2023-11-22
> **Reply**
>
> We thank the reviewer for the time and feedback. We address the questions below.
>
> > It would be interesting to see the results of SmoothECE on imbalanced (image) datasets such as SVHN.
>
> We agree that understanding the calibration of real networks in various settings is an important question (and imbalanced settings can be particularly interesting). However, we consider this out-of-scope for our paper, since our paper is about how to *define* calibration metrics, not about how the metric behaves on ML models of interest (though this is a natural direction for future work).
>
> Theoretically, our SmoothECE theorems do not assume anything about the distribution of predictions, and so will apply for any degree of imbalance. Empirically, the "Solar Flares" experiment (Fig 2b) shows an example of running our method on imbalanced data (the dataset has ~25% positive samples, ~75% negative samples). We can consider adding more imbalenced-data experiments if they would be informative.
>
> > The empirical results are missing experiments on more image datasets, especially that cover out-of-distribution scenarios, and different types of networks.
>
> See above; we consider this important but out-of-scope for our paper.
>
> > The overall contributions of the paper seem a bit limited as it boils down to applying (Gaussian) kernel to the original distribution for obtaining a smoothed estimate of it (which is already done in the prior work) and a technique to determine the kernel bandwidth.
>
> Note that our contribution is twofold: (1) deriving a theoretically-principled choice of Gaussian bandwidth, which was an unsolved complication in practice, and (2) proving that, with this bandwidth choice, the induced calibration measure satisfies the strong theoretical guarantees of a "consistent calibration measure".
> We acknowledge that the concept of Gaussian smoothing for reliability diagrams is not new (as we cite on page 3), but the details matter significantly, and it was not previously known that this method could satisfy meaningful theoretical guarantees.
>
> > It is a bit difficult to understand that, how the smooth reliability is visually more interpretable than the binned reliability.
>
> We have included a new section (Appendix E,  FIgure 3) in the updated pdf, which demonstrates how SmoothECE can behave *much* better than binned ECE in certain settings.
>
> > To what extent, this smoothECE can be trusted as a standalone miscalibration measure?
>
> The goal of our theoretical results is to demonstrate rigorous ways in which smoothECE "can be trusted as a standalone miscalibration measure." Specifically, what it means to be a "good miscalibration measure" was rigorously defined in (Błasiok et al 2023), who called such measures "consistent calibration measures." In this paper, we prove that SmoothECE is in fact a "good" calibration measure in the above rigorous sense – this is our main theoretical result.
>
> We hope this addresses all of your concerns. We are happy to discuss any remaining points further.

---

> > ### Comment · Reviewer_MdfL · 2023-11-22
> >
> > I thank authors for submitting responses to my comments. Most of the responses adequately address my comments, including results on imbalanced datasets, more results on non-image datasets, and the interpretation of smooth reliability. The point on novelty could have been explained better which also highlight the significance of contributions (1) and (2).

---

> > > ### Author Response · Authors · 2023-11-22
> > > **Reply**
> > >
> > > We thank the reviewer for acknowledging our responses, and we are glad most of the concerns have been adequately addressed.
> > > If the reviewer believes this work now meets the bar for ICLR, we respectfully ask they consider increasing score to an "Accept."

---

### Official Review · Reviewer_1zec · 2023-10-31

**Soundness:** 3 good
**Presentation:** 3 good
**Contribution:** 2 fair
**Rating:** 6
**Confidence:** 4

**Summary:**

The paper proposes a way to replace the usual reliability diagram with a smoothed reliability diagram, thus resolving some known issues with using binning schemes. The ECE implied by the proposed smoothed reliability diagram is a consistent calibration measure.

**Strengths:**

While a number of miscalibration measures that use kernels have been proposed, they are not "human-interpretable" like an l1-ECE is. In particular, they cannot be plotted on a reliability diagram. This paper thus fills an important gap. The proposed reliability diagram is also principled in that it leads to a consistent calibration measure, in the sense of Błasiok et al. (2023).

The paper is easy-to-follow, clearly describes the issue that is being targeted, and the proposed solution is well-contextualized with the latest literature.

**Weaknesses:**

In my opinion, the quality of the paper can be improved with more detailed research and better presentation. I have a number of questions, and I feel at least some of them should be answered before publication. (Thus I have given a contribution rating of 2 since I believe more evidence needs to be provided to show the proposed method indeed solves the problem satisfactorily.)

## Theory/method questions:
- Page 3 bottom (initial proposal for smECE): I feel that the actual smECE defined in eq. (4) makes more sense than \widetilde{smECE} de facto, so the \widetilde{smECE} is slightly distracting. Is there a reason for not just introducing the actual smECE earlier?
- What is the reason for picking the fixed point \sigma*?
- The discussion of Błasiok et al.'s work should be more detailed for someone unfamiliar with it. After Definition 5, I would expect some comments on why consistent calibration measures are a useful notion.
- Theorem 6: is this a good, bad, or ok "rate"? (perhaps in terms of the \alpha_1 and \alpha_2?) What does it mean in practice? Does the rate show up in experiments?
- Could you share intuition for Lemma 7 and Lemma 8?

## Experiment questions:
- The experimental analysis focuses on producing some reliability diagrams. These look nice, but is there a way to show the benefits of the consistency property that is highlighted?
- Can you show further evidence that the smECE is close to the true ECE in finite samples (on synthetic data)? Fig 1d is one simple setup.
- Is smECE as useful, less useful, more useful than the usual ECE for comparing the calibration of models?


## Minor comments:
- Page 3 top. "Thus we have a situation ... consistent calibration measures." I agree, and would further add that reliability diagrams are nice because they are interpretable by humans. Thus Widmann et al.'s kernel-ECE is very useful for comparison but its adoption has been limited due to the interpretability (aka, lack of reliability diagram) problem.
- Page 3 bottom \tilde{K} is used before being defined
- Page 5 (center): the methods would in fact apply to any of the binary-reduction-based calibration notions discussed in the paper "Top-label calibration and multiclass-to-binary reductions"

**Questions:**

## Theory/method questions:
- Page 3 bottom (initial proposal for smECE): I feel that the actual smECE defined in eq. (4) makes more sense than \widetilde{smECE} de facto, so the \widetilde{smECE} is slightly distracting. Is there a reason for not just introducing the actual smECE earlier?
- What is the reason for picking the fixed point \sigma*?
- Theorem 6: is this a good, bad, or ok "rate"? (perhaps in terms of the \alpha_1 and \alpha_2?) What does it mean in practice? Does the rate show up in experiments?
- Could you share intuition for Lemma 7 and Lemma 8?

## Experiment questions:
- The experimental analysis focuses on producing some reliability diagrams. These look nice, but is there a way to show the benefits of the consistency property that is highlighted?
- Is smECE as useful, less useful, more useful than the usual ECE for comparing the calibration of models?

---

> ### Author Response · Authors · 2023-11-21
> **Reply (part 1/2)**
>
> Thank you for your detailed review and suggestions. Below, we address each of your points (in the same order as the original bullet points).
>
> ## Theory/method questions:
> * Regarding distinction between $\widetilde{SmECE}$ and $SmECE$: we agree that indeed $SmECE$ is a slightly more natural notion as a calibration measure. Those two notions are close to each other, both in spirit, as well as formally (see for instance Lemma 10 in appendix A). On the other hand, the $\widetilde{SmECE}$ is more directly reflected in a visually interpretable reliability diagram, which we argue for constructing. As such, we chose to define both those notions and highlight the close relations between those two, while arguing for SmECE to be the more fundamental construction, which should be eventually used to report calibration error itself. A slightly more detailed discussion about this is present in Appendix A.
> * Choosing the fixed point $\sigma^*$ is to a small extent arbitrary. Lemmas 7 and 8 provide an upper and lower bounds of the distance to the nearest perfectly calibrated predictor in terms of $smECE_\sigma$ and $\sigma$. This leads to a range of parameters $\sigma$ for which those upper and lower bounds are good enough for the measure to satisfy the “consistency” requirement. One could, for instance, choose $\mu(D) := \min_{\sigma} (smECE_{\sigma}(D) + \sigma)$. Such a $\mu$ would also be a consistent calibration measure. Yet it wouldn’t necessarily provide a specific scale $\sigma$ for which $smECE_{\sigma}(D)$ is proportional to $\mu(D)$ — and the quantity related with $smECE_{sigma}$ is the one we are presenting on the reliability diagram. On the other hand, choosing a fixed point $\sigma_*$, provides an approximate minimizer for $smECE_{\sigma}(D) + \sigma$ (up to a factor of two, since $smECE_{\sigma}(D)$ is non-increasing with sigma), provides a scale on which $smECE_{\sigma}(D)$ (which is visualized on the diagram) is proportional to $smECE_{\sigma}(D) + \sigma$ (which is guaranteed to upper bound the distance to calibration), is easy to compute, and is in a sense “canonical” in a range of parameters sigma for which we would have the consistency property  — a choice that can be universally agreed on, to end up with a parameter-free calibration measure with respect to which different classifiers can be easily compared.
> * Thank you for pointing out that the importance of consistent calibration measures was not discussed enough, that is a great point. We have included additional discussion in the current write-up (Appendix E), which we will consider moving into the main body in the full version (where we will have more space).
> * The more precise upper and lower bounds between smECE and distance to the nearest perfectly calibrated predictor are given at the end of Section 3.3 (giving $\alpha_1 = 1, \alpha_2=1/2$). This provides a decent continuity guarantee for the smECE measure. We provided some synthetic examples in Figure 3, where this continuity is highlighted — binning ECE and the associated binning diagram behaves extremely differently on two very close datasets of (prediction, outcome), while smECE provides a stable picture of mis-calibration.
> * Regarding intuition behind Lemma 7 and Lemma 8. Lemma 7 is best understood in terms of duality theorem for distance to calibration: such a distance is (up to a constant factor) equal to maximum over all Lipschitz functions $w: [0,1] \to [-1,1]$ of a correlation between $w$ applied to the prediction $f$, and the residual between prediction $f$ and outcome $y$,
> i.e. $\sup_{w} \mathbb{E}[w(f) (f-y)] $
> (in this paper this latter quantity is called weak calibration error, see Definition 19 and Theorem 20 in Appendix D.1). With this in hand, it’s not that surprising why quantities like $(\widetilde{smECE}_\sigma + \sigma)$ upper bound the weak calibration error. For any given Lipschitz function $w$, we can bound the $\mathbb{E}[w(f)(f-y)]$ in terms of $ECE(\pi_R(f+Z), y)$ and $\sigma$ where $Z$ is a an independent gaussian with standard deviation sigma. Indeed, while extending the notion of smECE to non-trivial metrics on the space of predictions $[0,1]$, one of the main steps we had to take care of was generalization of the duality theorem (see Appendix B.1 and D.8). Lemma 8 follows from observing that smECE(D) is Lipshitz-continuous with respect to the Wasserstein metric on distribution D (see Lemma 23 for a formal statement of this). Then, by comparing smECE(D) to smECE(D'), where D' is the closest perfectly-calibrated distribution to D, we obtain the bound in Lemma 8.

---

> > ### Author Response · Authors · 2023-11-21
> > **Reply (part 2/2)**
> >
> > ## Experiment questions:
> > * Indeed, that is a great point. We have added a set of synthetic reliability plots highlighting how commonly used binning variants of ECE behave extremely unreasonably (on a carefully crafted data) due to discontinuity issues, and how our method provides much more sensible diagrams in this exact scenario. These are included in the new Appendix E.
> > * The current experimental section shows some examples on which SmoothECE happens to behave closely to binning ECE on actual data. In contrast, and more importantly, we have now added synthetic examples for which SmoothECE behaves drastically differently than binning ECE (Appendix E). We argue that in those scenarios it is the binning ECE that behaves in a very undesirable way, and SmoothECE is fixing inherent problems with this definition, while being accompanied by informative reliability diagram.
> > * See above
> >
> > ## Minor comments:
> > * Thank you. Yes, that was exactly the point why we have introduced the SmECE: ECE by itself is mathematically poorly behaved, and better methods for stand-alone measuring calibration error has been proposed (as well as abstracting away the desirable properties of such measures, which ECE fails to posses). Unfortunately, those metrics were not widely adapted, in part because binning ECE can be accompanied with a useful reliability diagram. We proposed a method which has desirable mathematical properties of those better calibration measures, while being associated with a visually interpretable calibration diagram.
> > * Thank you, fixed.
> > * Indeed, thank you for pointing us to this work. We have added this citation in the revised pdf.
> >
> >
> > **Conclusion:**
> > We hope this response addresses your concerns, and we are happy to discuss any remaining concerns further.

---

### Official Review · Reviewer_SRwZ · 2023-11-01

**Soundness:** 3 good
**Presentation:** 3 good
**Contribution:** 3 good
**Rating:** 6
**Confidence:** 3

**Summary:**

The paper proposes a novel calibration metric called SmoothECE based on the use of kernels.
Compared with the commonly used ECE, which is also referred as BinnedECE, it does not suffer from discontinuity.
More importantly, it is a consistent calibration measure and comes with a reliability diagram for visualization (unlike some other proposed metrics).
Even though SmoothECE relies on the use of Gaussian kernels, it has a procedure to choose the kernel bandwidth, which makes it hyper-parameter free.
Calibration of classifiers trained on common benchmarks are assessed by SmoothECE and compared with BinnedECE, showing that it performs similar to BinnedECE and is easy to visualize.
A Python package is developed for this method to use.

**Strengths:**

**originality** The proposed SmoothECE is novel so do the theoretical results.

**quality** The proposed method is sound. It's consistency is proved as a result of the combination of the use of reflected Gaussian kernels and the way to set $\sigma$, which is very neat.

**clarity** The paper is well-written and easy to follow.

**significance** SmoothECE is a drop-in replacement of BinnedECE and can be potentially widely used by the community. Apart from this, as SmoothECE also alleviates the discontinuity problem of BinnedECE (to be confirmed in Questions), it can enable more work that rely on differentiating through calibration metrics, which is very prohibited by BinnedECE.

**Weaknesses:**

The experiment section is weak.
If the proposed method alleviates the discontinuity problem of BinnedECE (to be confirmed in Questions), some experiments showing how it can be beneficial (e.g. optimizing a loss involving the calibration metric) should be included.

The code is not provided.
Perhaps the author(s) can use https://anonymous.4open.science to share it anonymously.

**Questions:**

Can you clarify if SmECE is differentiable?
If so, can we include it in the loss while training classifiers?

In figure 2, what are the shaded areas for the smooth reliability diagrams?
I thought they are kernel density estimates of the predictions but I don't understand why for (d), the red line is surrounded by that area.

---

> ### Author Response · Authors · 2023-11-21
> **Reply**
>
> We thank the reviewer for their time and feedback. We have implemented several of your suggestions, and we respond to your questions below.
>
> * Re. discontinuity problem: Thank you for your suggestion – it will surely improve our paper to provide experimental data highlighting the issues with discontinuity of BinnedECE and associated reliability diagrams, as well as showing how those problems disappear when using our method instead. We have included such a diagram now, in Appendix E (Figure 3) of the updated pdf.
>
> * Re. code: We have now uploaded the anonymous code here: https://anonymous.4open.science/r/smoothece-anon/ (also in the attached .zip file), as we mention in our common response.
>
> * Thank you for pointing out the inconsistency in Figure 2, plot (d). The shaded areas (and the thickness of the red line) indeed were meant to indicate the kernel density estimates. The plot (d) was produced by using our library with a different option enabled, showing the confidence intervals for the red line. We will change this back to the standard format, so that it is consistent with other plots. (We chose not to focus on confidence intervals in this paper in order to provide cleaner plots, but they are an option supported by our library).
>
> * Re. differentiability: Yes, on any fixed scale $\sigma$, the $SmoothECE_{\sigma}$ is differentiable. Thus it is possible to include it in the loss while training classifiers. Our current library is written in numpy, but it should be easy to port to pyTorch/JAX, and perform the differentiation using the standard autodiff. This is an interesting idea for future work.
>
> We hope this addresses the reviewer's concerns, and we are happy to discuss any remaining issues further.

---

> > ### Comment · Reviewer_SRwZ · 2023-11-22
> >
> > Thanks the authors for their reply. I acknowledge that I've read the response.
> > The new result on showing SmoothECE is much more robust than BinnedECE is nice.
> > It's unfortunate that the current NumPy-based implementation cannot produce a differentiable implementation of SmoothECE out of the box, even though it's mathematically differentiable.
> > I highly encourage the authors to work on that as I can see that can be something highly impactful.

---

### Official Review · Reviewer_PxLa · 2023-11-01

**Soundness:** 3 good
**Presentation:** 3 good
**Contribution:** 3 good
**Rating:** 8
**Confidence:** 3

**Summary:**

This paper proposed the smooth ECE, which uses kernel smoothing to compute the Expected Calibration Error(ECE) estimator. The proposed estimator is naturally related to a smoothed reliability diagram. The estimator is shown to be consistent and computationally efficient and sample-efficient.

**Strengths:**

The paper is well-written and easy to follow. It provides a nice alternative to the commonly used binned ECE estimator, which has the potential to be widely used in the model calibration literature. The theoretical property of the proposed estimator is carefully studied and the computational complexity is also addressed.

**Weaknesses:**

1. The major contribution of the paper is the new smooth ECE estimator over the commonly used Binned ECE. The paper discussed several disadvantages or flaws of the binned ECE in the introduction, but I think these flaws are not well demonstrated in the experiments, e.g. "changing the predictor by an infinitesimally small amount may change its ECE drastically", "overly sensitive to the choice of bin widths." I think it is beneficial to include some synthetic experiments to demonstrate these problems of the Binned ECE estimator and show how the proposed estimator overcomes them.

2. One property of the proposed estimator is consistent in the sense of (Blasiok 2023). It is different from statistical consistency, I think a bit more discussion on why this property is desirable is helpful.

**Questions:**

1. The authors emphasize at the end of section 2 that ”consistent calibration measure does not refer to the concept of statistical consistency“. Can the authors comment a bit on the statistical consistency properties of the proposed estimator? e.g. convergence to the ECE as sample size increases?

2. On top of page 6, the reflected Gaussian kernel uses sum over $\pi_{R}^{-1}(y)$, is this an infinite sum by the definition of $\pi_{R}^{-1}(y)$? How is it computed in practice?

Minor question

3. In Deep Network paragraph of Section 4, it says "ResNet32", but it is "ResNet34" in Figure 2.

---

> ### Author Response · Authors · 2023-11-21
> **Reply**
>
> We thank the reviewer for their time and feedback, and we are glad the reviewer appreciates the contribution of this work.
> We address the questions and weaknesses below.
>
> > I think it is beneficial to include some synthetic experiments to demonstrate these problems of the Binned ECE estimator and show how the proposed estimator overcomes them.
>
> Thank you for this suggestion. We have now updated the pdf with a new section in Appendix E, including a new Figure (3), to help illustrate the problems with Binned ECE, and the resolution of Smooth ECE. We hope this addresses your concern.
>
>
> > Can the authors comment a bit on the statistical consistency properties of the proposed estimator?
>
> Yes, we have now added an explicit discussion of this in Appendix E.1, which we will reproduce here:
> Our estimator of SmoothECE also satisfies the classical criteria of statistical consistency.
> Specifically Theorem 9 shows that as the number of samples $m \to \infty$, the finite-sample estimation error $\varepsilon \to 0$. (In fact, Theorem 9 is stronger than just asymptotic consistency--- it provides a quantitative generalization bound).
>
>
> > On top of page 6, the reflected Gaussian kernel uses sum over is this an infinite sum by the definition of? How is it computed in practice?
>
> Appendix C discusses how the SmoothECE can be easily and efficiently computed. In short, the infinite sum in the definition can be approximated up to an error of $\delta$ by using only $O(\sqrt{\log \delta^{-1}})$ leading terms (in practice: just three such terms are enough). Moreover, even this truncation does not have to be computed explicitly: to approximately convolve function $r :[0,1] \to \mathbb{R}$ with the “reflected gaussian kernel” on [0,1], one can extend it by reflecting around 0 and 1 to residual $\tilde{r} : [-1, 2] \to \mathbb{R}$ (or, more formally $\tilde{r} : [-T, T+1] \to \mathbb{R}$ for some $T\approx \sqrt{\log \delta^{-1}}$), apply FFT to compute convolution with a standard gaussian kernel of such an extended function, and truncate this convolution to interval $[0,1]$.
> This "reflection+convolution trick" is what we actually use in the code implementation, which is now attached as supplementary material.

---

> > ### Comment · Reviewer_PxLa · 2023-11-22
> >
> > I appreciate the authors for the detailed response and additional experiments. The new section in Appendix E highlights the advantage of the proposed method over binned ECE. I improved my rating.

---

### Author Response · Authors · 2023-11-21
**Common Response to All Reviewers**

We thank all reviewers for their time. We wish to highlight several updates we have made, and we will respond to each reviewer individually in separate threads.

* Code release: We have uploaded our code as an anonymized python package here:
https://anonymous.4open.science/r/smoothece-anon/ . (And also attached it as a .zip to OpenReview). This includes a "pip-installable" package which computes smECE and plots the reliability diagrams, along with notebooks to reproduce the experiments in our paper.

* Added new expository section (Appendix E), including a new figure (Figure 3), to help illustrate the discontinuity problem of ECE, and the continuity of our proposed solution (SmoothECE).

---

### Meta-Review · Area_Chair_GP7b · 2023-12-12

**Metareview:**

The paper offers a novel tool for checking reliability and calibration of predictors, with well-founded theoretical backing and usefulness in practice.

**Justification For Why Not Higher Score:**

The method could potentially be of limited audience appeal

**Justification For Why Not Lower Score:**

The paper makes a solid technical and practical contribution

---

### Decision · Program_Chairs · 2024-01-16

Accept (poster)